# Impacts of large-scale deployment of vertical bifacial photovoltaics on European electricity market dynamics

Laszlo Szabo[1], Magda Moner- Girona [2], Arnulf Jäger-Waldau [3], Ioannis Kougias [4], Andras Mezosi[1], Fernando Fahl [5] & Sandor Szabo [3] ✉

Self-sufficiency, climate change and increasing geopolitical risks have driven energy policies to make renewable energy sources dominant in the power production portfolios. The initial boom in the mid-2000s of global photovoltaic installations demonstrated the feasibility of the ambitious renewable energy targets. However, this rapid scale-up has introduced challenges, including price volatility and system integration issues. This communication calls the attention to these emerging challenges and offers quantitative insights on how rapid adoption of a more diversified photovoltaics deployment strategies can mitigate price volatilities, reduce fossil fuel dependence and steer Europe towards a forward-thinking sustainable energy pathway. The analysis reveals that as innovative bifacial photovoltaic systems are incorporated on a large-scale disruptive scenario, four main patterns emerge: economic value of solar production increases, base-load electricity price decreases, sun-rich countries expand their solar contributions, whereas nations with ample grid interconnections enhance their energy imports from neighbouring countries. It also underscores the importance of maintaining photovoltaics an attractive option for energy investors and traders in the future. Establishing this groundwork is critical since a successful integration of large-scale solar systems contributing to decrease price volatilities in Europe and US will carry significant repercussions for global energy policy formulation.

To increase energy independence and accelerate the transition to climate neutrality the most recent global and European climate targets for 2040 request countries all over the world to scale-up clean energy investments. The European Commission's Solar Strategy Communication[1] of 2022 calls for about 450 GW (AC current) of PV system capacity additions between 2021 and 2030 (Given the current trend of installing 1.25 to 1.3 times the AC capacity in DC, this would bring the total nominal photovoltaic (PV) capacity in the European Union (EU) approximately at 720 GWp). To reach the increased ambition targeted by the REPowerEU plan[2], deployment of RES and PV must be accelerated. The target of reaching 720 GWp by 2030 is particularly ambitious when compared to the current installed capacity of 268 GWp[3].

Nevertheless, achieving such ambition targets appears to increase further the current challenges. The electricity grids are ageing all over the world: in Europe and US one third of the assets are more than 40 years old and this share in the European Union (EU) will exceed 50% by 2030 in ref. 4. This poses a great challenge to integrate more

[1]Regional Centre for Energy Policy Research, Corvinus University of Budapest, Budapest, Hungary. [2]Renewable and Appropriate Energy Laboratory, University of California, Berkeley, USA. [3]Joint Research Centre, European Commission, Ispra, Italy. [4]Elpedison SA, Athens, Greece. [5]European Dynamics Luxembourg S.A., Luxembourg, Luxembourg. ✉e-mail: sandor.szabo@ec.europa.eu

Renewable Energy Sources (RES), but inevitable grid investments will lead to a smarter, more digitalised system that will enable large-scale RES integration in the long term. A recent study[5] calculated a conservative benchmark of over 1 TWp for the PV capacity potential in the EU on rooftops (560 GWp), vertical bifacial PV along roads and rails (403 GWp) and floating PV (157 GWp) on reservoirs (which can also be bifacial). Building-integrated PV (BIPV) and agrivoltaics[6] create even bigger potentials for bifacial PV applications, facilitating the development of net-zero buildings and enabling the dual use of land, respectively. As the future portfolio is expected to be a distribution of a mixture of optimal and vertical bifacial PV deployment with transient orientations following the available infrastructure (e.g. highways, BIPV, agrivoltaics), in the modelled scenarios different levels of vertical bifacial PV deployment were modelled.

Many US states and local governments have set specific targets for solar and wind energy capacity installations. For example, California has set a goal of achieving 100% carbon-free electricity by 2045 leading to 148 GW of renewable energy buildout in two decades[7]. By incentivizing rooftop solar installations, investing in large-scale solar farms, and implementing supportive policies, California aims to harness solar PV as a primary source of renewable energy.

Decarbonisation plans gave global momentum to solar energy to become a prevalent electricity source[8]. However, stakeholder groups have voiced quite diverging visions on whether solar PV should be the dominant component of the future electricity generation portfolios. While there is a general consensus that achieving clean and secure energy systems requires rapid and extensive deployment of PV[9] and that renewable technologies are ready to scale up to multi-TW levels[10], and there are hundreds MW size bifacial PV projects realised worldwide[11], there are differing perspectives on the role PV should play. Some groups highlight that the primary threat to reach carbon neutrality in the energy sector is posed by stop-and-go policies that slow the deployment of RES[12]. These perspectives also draw attention to the negative prices experienced in the EU and USA wholesale power markets in recent years. They point out that conventional PV systems, which generate electricity mostly in the hours around noon, put downward pressure on wholesale prices. If the current trend of installing usual PV systems continues, the value of the PV electricity sold in the wholesale markets could reach zero or even negative levels with increasing frequency. This devaluation of the economic value of solar electricity imposes risks on further PV deployment, the establishment of Power Purchase Agreements (PPAs) with large consumers, and on the deployment of merchant solar systems that do not require public financial support.

Moreover, projections for negative prices introduce insecurity for both regulatory bodies and investors, hindering the scale-up of the required investment needed for climate change mitigation strategies. These strategies cannot afford to lose momentum in the transition to clean energy, especially in counties/regions leading the deployment of solar and wind such as Australia, China, EU, and the USA. A slowdown would have detrimental effects on the global value chains also affecting the rest of the world.

Several studies have addressed the effects of price signals in either stimulating or hindering new investments[13]. A recent US analysis emphasised that these price signals provide investors important location-specific information to where (nodes or locations) PV deployment can create more value[14,15]. A spatial analysis for Germany concluded that the strong fiscal support for RES combined with marginal cost bidding has led the more frequent occurrences of negative prices in the day-ahead markets. The analysis also identifies demand-side markets (e.g. green hydrogen production, battery storage, and load shifting) as potential solutions to address market dysfunctions[15]. Another study, based on statistical analysis of the German market, revealed that renewables, with their low marginal costs and subsidised remuneration, had a strong price suppression effect on the spot power markets. Consequently, fossil power plants experienced reduced profit margins and lacked incentives to invest[16].

It is evident that there is a pressing need for a significant shift in financial conditions[17] to incorporate the risk of cost overruns[18] and risk-adjusted costs of RES[19]. Simultaneously, the PV deployment must embrace disruptive patterns as opposed to standard approaches. Over the next few decades, society will face dramatic changes as climate change and the electrification of several sectors (i.e. transport, heating, and cooling, industry) will lead to a significant rise in power demand[20]. With the simultaneous rise of energy demand, energy returns on investment need to be taken into account for renewable and storage technologies[21]. The number of prosumers will increase leading to a profound transformation of the utility sector[22]. Alternative options to mitigate the integration hurdles and low PV electricity value rely either on storage options and emerging technologies that convert renewable electricity into chemical energy carriers, a concept known as Power-to-X (P2X). Battery storage is a promising option that has recently gained momentum as reflected in the successful competitive tenders to deploy utility-scale battery energy storage systems (BESS). Still, the selected systems typically refer to 2 to 4-hour duration BESS, which can only partially transfer the excess solar generation to the evening peak, especially when the PV deployment reaches the set targets[23]. Long-duration storage technologies have not reached yet the market maturity required to achieve economy of scale and support the integration of large capacities of RES, at least not in the short term. Accordingly, system stability requires long-duration storage, which presently lacks market readiness. Concerning P2X utilisation of excess solar PV generation, these options require interplay with other markets (industry, electromobility, etc.) that makes their implementation less efficient and more complicated/time consuming.

This study considers the potential of novel deployment practices for PV to quickly reshape the European electricity sector. The proposed deployment options, such as vertical bifacial PV, create added value of the PV production to extend the production time of PV electricity to periods where it is more valuable to the consumers and easier to be dispatched. From system integration point of view, vertical East-West facing PV systems offer bigger potential compared to inclined South facing PV. Such setup typically produces 30% in the 3 midday hours in contrast to the South facing PV that produce nearly 70% of its output in the midday (calculations based on measurements[24] and PVGIS[25]). This difference is higher than the one between off-shore and on-shore wind systems, indicating the potential role vertical systems can play. Vertical bifacial PV installations offer a fast, no regret option that decreases land use competition and mitigates PV system integration issues. Bifacial module requires negligible added cost compared to usual PV modules (by replacing the back cover to transparent one) extending the range of PV applications to options that minimise land use changes and can produce electricity in a strikingly different pattern.

## Results
### Reshaping the energy landscape

Achieving the global target established at COP28 to triple renewable power capacity by 2030 significantly relies on creating favourable conditions for this expansion. Over the past decade, the reshaping of the energy landscape was characterised by massive investments, particularly in the solar PV and wind-based electricity generation technologies. While PV deployment scaled up in unprecedented volumes, future growth scenarios are even more ambitious[26]. To date, increased PV generation has helped alleviate the mid-day peak demand. However, accelerating PV deployment to fast-track the transition to climate neutrality[2,27] faces rising challenges. One challenge arises from the prevailing solar PV business approach, which focuses on maximising the total generation by designing PV systems with "optimal"

orientation and tilt. This concentration of production around midday creates system integration, technical, and market problems:

i. High PV-based electricity generation that exceeds demand creates challenges in the transmission/distribution network operation. So far, integrating the solar PV output in the power system has been feasible in regions such as the EU, US, and China, primarily due to their ability to integrate the relatively low PV capacities within the existing transmission and distribution networks. However, several regions (including several EU Member States, states in USA, regions in China and Australia) face or will soon face solar output exceeding demand at noon.

ii. Despite advances in supporting RES integration such as a more precise forecasting of PV systems' generation and remote control of their operation, production imbalances remain and exacerbate with increased PV capacities.

iii. Furthermore, given that the technological and market maturity of storage and its regulatory framework are still taking shape, above a certain level of installed capacity, transmission system operators (TSOs) need to intervene and implement stringent measures during periods of high RES output. These measures include curtailing excess solar energy generation, adjusting dispatch schedules, activation of demand response mechanisms, and imports' restrictions. Such interventions carry significant economic repercussions such as substantial network expansion costs, distort the operation of the internal electricity markets, and hinder efforts to decarbonise energy systems[28].

iv. Additionally, connection agreements for small- and medium-scale PV systems at the distribution level are often delayed or denied due to local capacity constraints faced by the distribution system operators (DSOs)[29].

v. Wholesale market price cannibalisation occurs as the growing integration of renewable energy sources with zero marginal costs leads to devaluation of their market worth thereby diminishing investment incentives in solar PV[30]. This effect becomes more pronounced with the continued expansion of PV deployment, pushing sunny time periods wholesale prices to zero values[15] or even negative values[16]. Consequently, the presence of low-value power output threatens the economic viability of merchant PV systems and imposes a significant risk factor when signing into RES PPAs[31].

Continuing to focus on uniform installations of "optimal-oriented" PV arrays would exacerbate all five challenges and would undermine future revenue generation by depressing midday wholesale prices to zero or negative range, making the required future development of merchant PV systems less feasible, and subsequently imposing significant economic burden on consumers.

Financing PV systems is an important aspect of achieving large scale deployment of solar systems, as they are very capital-intensive investments. Financing depends on the type of investment, in which PV systems can generally be categorised depending on size and application (residential and commercial). See financial options for Small Systems, Medium Systems and Commercial-scale systems in the Financing Scheme chapter below. The concept of vertical bifacial PV of this paper, mainly refers to medium and commercial systems.

## Alternative deployment options in the EU

The target of reaching 720 GWp by 2030 is particularly ambitious when compared to the current installed capacity of 268 GWp. Some analysts even anticipate that with the growing adoption of electromobility, heat pumps for less carbon-intensive space heating, and green hydrogen for industrial processes, the demand for PV in the EU could even exceed 1 TW by 2030 in ref. 32.

In the long-term, several promising options support the deployment of solar PV: investments in grid infrastructure and demand-side

response (DSR), storage (standalone batteries or hybrid RES systems) link the production with other demand sectors and market mechanisms enhancing system adequacy.

Optimised Grid Infrastructure Investments: Prior to extensive grid expansion or upgrades, it is recommended to streamline the design of PV and RES systems. This approach helps minimise the capital-intensive nature of grid investments, as highlighted by ENTSO-E's estimate of EUR 600 billion[33].

Integration with Demand-side Response (DSR) and Electro-mobility: DSR presents significant potential to complement solar PV deployment by enhancing grid stability and maximising the value of solar energy. Through DSR technologies, consumers can adjust their electricity usage patterns to align with solar generation peaks, thus increasing the value of PV-produced electricity. Large European industries have already implemented this approach, often participating in capacity markets.

Linking Production with New Demand Sectors (Sector Coupling): Exploring links between PV production and emerging flexible demand sectors, such as electrolysis for green hydrogen (P2X), is a promising option[34]. However, widespread implementation of this concept faces challenges, with current capacity falling short of ambitious targets set by initiatives like the EU Hydrogen Strategy with an electrolyser manufacturing capacity of 3.9 GW by late 2023 compared to the 17.5 GW 2025 target. Parallel approaches are needed to address PV deployment challenges until P2X technologies scale up.

Market-based Frameworks for Long-term Sustainability: Long-term measures should aim to establish market-based frameworks for RES systems to operate without public subsidies to foster long-term sustainability and innovation. Apart from societal costs, public subsidies distort market dynamics, leading to inefficient allocation of resources and hindering innovation by shielding renewable energy providers from market pressures. While Contracts for Difference (CfDs) can support PV investments by sharing risk[35], they require public support, distort market dynamics, and fail to address low-value issues during peak-production times.

Renewable Power Purchase Agreements (PPAs): PPAs are contracts between producers and consumers, often facilitated through intermediate electricity suppliers. These agreements represent a priority mechanism for the EU (revised Renewable Energy Directive 2023/2413[36]) in its efforts to facilitate the supply of clean and affordable power while reducing reliance on energy imports. PPAs can incorporate CfDs in commercial terms without introducing market distortions. However, the uptake of PPAs, especially for solar PV, has been slower than anticipated due to investor concerns about the future low value of produced energy.

Capacity Mechanisms in Redesigned Electricity Markets: Capacity mechanisms in the redesigned electricity market offer significant advantages to secure system adequacy and, simultaneously, support investment in power generation. The recent reform[37] of the market design makes capacity mechanisms a permanent feature of the market, and such market integration minimises potential distortive effects. However, the potential benefits that capacity mechanisms can bring to PV investment remain modest given the technology's limited capacity to ensure supply adequacy and secure sufficient revenue in capacity tenders.

In the short term, it is crucial to identify quick, no-regret options capable of accommodating the unparalleled scale-up penetration of PV technology. The present paper explores such a no-regret option: the potential benefits of installing large quantities of PV systems in a more diversified and sustainable manner. This strategy capitalises on several factors, including the adoption of new module technologies like bifacial modules, and innovative installation practices such as non-standard orientation, vertical PV, and tracking systems[24]. Research indicates that vertical PV systems equipped with bifacial modules can generate up to 15% more electricity than conventional systems[38].

Vertical installation also addresses the sensitive issue of limited land availability for PV deployments, enabling the utilisation of a broader range of areas, including applications in agricultural land and greenhouses, a concept also known as agrivoltaics[6]. Additionally, these systems can be integrated into linear infrastructures like highways[39,40] and incorporated into building structures as building-integrated PV (BIPV) solutions.

To further advance the development of PV systems at even higher shares, it requires solutions that can effectively manage solar output variations and alleviate the growing challenges associated with mounting balancing and integration. While storage-based solutions are expected to play a significant role in the long term, this paper highlights the opportunities presented by emerging PV technologies and innovative system designs to minimise the overall costs of system transformation. It presents a model-based approach aimed at illustrating the diverse impacts of deploying high shares of diversified solar PV installation.

## Financing schemes for different types of solar PV systems

Solar PV systems can generally be categorised depending on size and application (residential and commercial). Each type may have different financing and contractual considerations:

**Small systems.** These are typically designed to power individual homes or small businesses. Financing for small systems often involves specific residential loans, net-metering and net-billing options (with physical or virtual connection). In that sense small systems may sign contracts with solar installers, financing companies, or utility companies to establish net-metering or net-billing arrangements.

**Medium systems.** Medium-sized systems are typically larger than residential systems and often serve small to mid-sized businesses or community installations. Financing options for medium systems may include commercial loans or PPAs tailored to the specific needs. Such agreements may involve negotiations with commercial financing entities to secure various services, including solar PV system installations, energy efficiency measures, storage and energy management, and maintenance services. This broad range of energy solutions and financing terms are provided by energy service companies (ESCOs).

**Commercial systems.** Commercial-scale systems are designed to meet the energy needs of larger businesses, industrial facilities, or institutions. Financing for commercial systems may involve commercial loans, leases, third-party ownership models, or direct investment by the business.

The concept of vertical bifacial PV in this paper mainly refers to medium and commercial systems, including the case of community energy systems. Each type of system has its own considerations regarding financing, contractual agreements, and regulatory requirements, tailored to the size, application, and specific needs of the customer or entity installing the solar PV system.

## Modelling the impacts of vertical bifacial PV on the European Power Market

The present analysis employs the European Power Market Model (EPMM)[41], which is a unit commitment dispatch optimisation model. Recent applications of EPMM address various energy policy issues relevant to the RES deployment, such as cross-border RES exchange in the Central and South Eastern Europe energy connectivity (CESEC) countries[42], and the impact of carbon taxation tools like the carbon border adjustment mechanism (CBAM) on RES deployment[43] with rich representation of technological progress[44]. The primary model objective is to satisfy the electricity consumption needs at the lowest system cost, considering the characteristics of available power plants and cross-border transmission capacities in the European power system.

The modelling process minimises electricity demand costs, encompassing factors like start-up and shut-down costs, production costs (fuel and $CO_2$ emissions), and RES curtailment. EPMM adopts a bottom-up approach, covering both conventional and renewable generation and, as well as energy storage. The grid representation employs a simplified Net Transfer Capacity approach, enabling it to capture commercial electricity trade among the covered European countries. There are a number European-level models for analysing energy market development (PRIMES[45] AERTELYS[46], POTENCIA[47]) but the EPMM was selected in this innovative PV deployment assessment as it captures the required hourly resolution and auxiliary market details for the modelling. The model simultaneously optimises operations for all 168 hours of a typical week, to determine power plant operations and their production levels. It runs for each week of a given year, considering a representative weather pattern that includes wind and solar irradiation data. EPMM is capable of endogenously modelling 41 electricity markets in 38 countries across the European Network of TSO for Electricity, ENTSO-E, network, providing a comprehensive view of the European electricity landscape.

EPMM fits within the broader framework of Operation Decision Support tools[48,49] that classifies various energy and power sector models. It adopts a bottom-up approach, covering both conventional and renewable generation as well as energy storage.

It is important to highlight, that the present analysis exclusively focuses on grid-connected PV systems and does not encompass PV capacities dedicated to hydrogen production and industrial processes. The reason for this exclusion is the considerable uncertainty whether these PV installations will be connected to the grid or operated as dedicated stand-alone systems. Figure 1 gives a simplified depiction of the model's input and output components.

The analysis in this paper presents a limited comparison of the impact of the modelled higher penetration of bifacial PV systems, assuming that all other important variables remain unaffected. This means the other flexibility options (e.g. increasing transfer capacities between countries, higher level of storage and demand side options) are kept at their reference pathway, which means the dynamic interactions with these options are kept limited in the analysis. This is a strong assumption, as the modelling is applied for the next 15 years. As these other options would affect wholesale prices and market values of production, the comparison remains rather static, but it still indicates the range of potential contribution of the bifacial technology to the power sector transformation.

## Modelling framework for baseline scenarios

The European Power Market Model covers the interconnected European Network of Transmission System Operators for Electricity and neighbouring countries characterised by significant energy trade. This coverage includes the EU 27 member states, the United Kingdom (UK), Switzerland, the six Western Balkan countries (WB6), Turkey, Ukraine, Moldova, Norway and Belarus.

Fuel prices play a pivotal role in shaping the generation mix, determining the competitiveness of the fossil-based generation plants, such as coal and natural gas-based units, in contrast to modern RES. In addition, the $CO_2$ price of the Emission Trading Scheme (ETS) is an important driver over the composition of the generation mix within the EU 27. It is worth noting that neighbouring countries trading with EU Member States benefit from tax advantages, as they are exempt from paying the ETS carbon price. This exemption leads into a more carbon-intensive generation mix in the bordering countries.

Exogenous coal and Title Transfer Facility (TTF) gas prices within the EU, alongside the trajectory of ETS carbon pricing were used in the assessed scenarios. It should be acknowledged that gas and coal prices

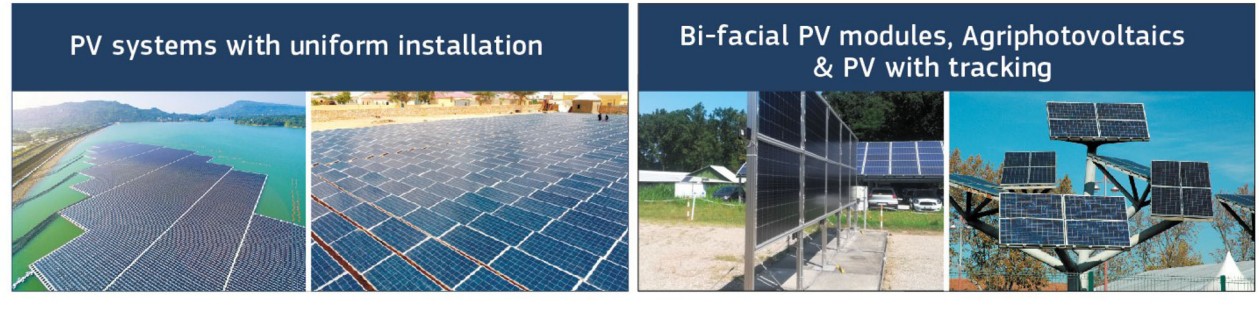

**Fig. 1 | Simplified scheme of the model's input and output components.** Comparative analysis involving two different sets of input data related to PV production across the European countries: the uniform installation (100% optimal monofacial) and the alternative option (diverse orientation / bifacial). The method incorporates various portfolio compositions, examining scenarios incorporating a spectrum of PV production levels, spanning from 0% to 50% inclusion within these portfolios. Subsequently, the model identifies new equilibriums within the energy markets, revealing the consequential effects on several key aspects. These include alterations in electricity generation portfolios, influences on wholesale pricing dynamics, PV output curtailments tendencies, and induced substitutions in fuel usage and reductions in $CO_2$ emissions.

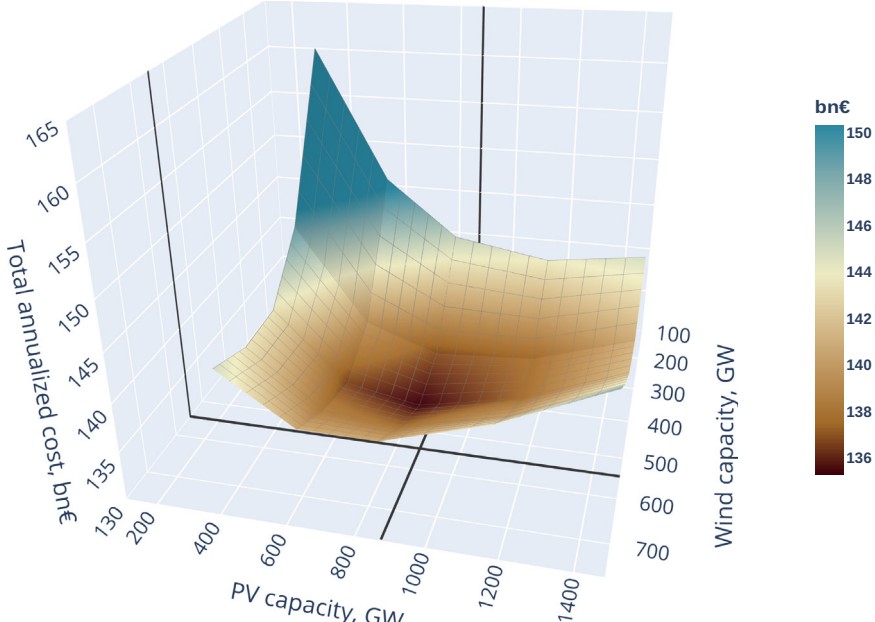

**Fig. 2 | Total annualised system cost range at various PV and wind capacity levels, 2040.**

differ country by country within the model, reflecting the varying transportation costs associated with these commodities.

Various sources were assessed and used for the modelling: e.g. Gas price (EGMM modelling, ICE INDEX[50]), Coal price (World Bank[51]; and IEA WEO[52]), $CO_2$ price (IEA WEO[52] and EU[53,54]), and their exact values are included in Supplementary Data 1–5.

There are two basic options to determine the installed renewable capacities in the EPMM model. It can either use its endogenous investment module for planning capacity expansion, based on minimised overall energy system costs to satisfy the projected demand growth, or renewable capacity deployment could be based on the official national trajectories outlined in the National Energy and Climate Plans (NECPs). The endogenous investment module was used in the analysed scenarios, as the NECPs show low ambitions presently compared to the EU targets. This would give a static snapshot of the current politically driven decision, while the endogenous investment module, with the given range, provides a more economically rational decision, based on the assumed dynamic development of capital costs. However, the endogenous investment module has its uncertainties as shown later in this section. A visual representation of these installed capacities and their total cost ranges can be found in Fig. 2 and Fig. 3. Total costs include the annualised investment cost of new capacities and the short-term operational costs, such as fuel, $CO_2$ and O&M costs.

The figure illustrates that the cost curve is relatively flat in a substantial range of variable Renewable Energy Sources (vRES) capacities in 2040 reaching minimum at 875 GW PV and 539 GW wind generation. Therefore, detailed results are shown in the following sections for three sets of PV capacities, staying at the central PV value, and also for a +− 2% total cost range, showing the uncertainty in the estimates. This gives a lower estimate of 649 GW and a higher estimate of 1178 GW PV capacity range, providing the range of uncertainties in the modelled investment decision and it is depicted in the following figure. Figure 2 demonstrates that the steepest cost increase happens, when both wind and PV investments are at low level, suggesting that sluggish investment in vRES could prevent reaching a cost optimal development in the EU power sector.

### Simulation effects of varying vertical bifacial PV deployment

Several sensitivity scenarios were introduced, each representing variations to the base scenarios. These variations involve an increased

share of vertical bifacial PV installations. It is important to note that, in the base case, no investments in vertical bifacial PV module are assumed, implying that all PV installations are the typical inclined south-facing PV type. In contrast, the five sensitivity scenarios were modelled with increasing shares of vertical bifacial panels, ranging from zero to 50% shares in 10% increments by the year 2040. It is worth emphasising that, across all modelled scenarios, wind generation is kept on a consistent growth pathway, with its share expected to rise to 539 GW by 2040. Figure 4 illustrates alterations in the generation mix across scenarios when including PV shares of 0 and 50% deployment of vertical bifacial PV technology.

In the context of the energy transition process, an important policy question is to what extent conventional generation, including gas, coal and nuclear, is impacted by the integration of a more diversified PV portfolio, including both standard and vertical bifacial modules. When examining the transformation of the overall electricity mix in the EU, varying impacts are observable based on the different capacities of vertical PV installations. The model output clearly shows an increase in solar generation by 2% and 3.6% in 2030 and 2040, respectively (Reference PV scenario). This increase exceeds 5.3% in the high PV scenario, clearly showing the potential of the vertical system. Notably, a substantial increase in the electricity injected into the grid is evident with higher vertical PV utilisation, primarily replacing gas and nuclear generation. Figure 4 shows that while the variations in the overall power production levels among scenarios and deployment levels remain relatively minor, the impact of the *high PV* deployment scenario on conventional technologies, particularly gas and nuclear, is substantial. The "50/50" setup leads to nearly 12% gas substitution as a result of different orientation and this is clearly remarkable as a no regret option not entailing significant system cost increase (within the +−2% range) in terms of supply security, greenhouse gas emissions, cost, and price fluctuations. The sizeable 12% reduction in gas consumption attributed to different orientations is definitely remarkable and represents a promising no-regret option. The main reason for having only a minor coal replacement effect can be attributed to the fact the that coal is already phased out almost completely in the system by 2040. Furthermore, some replacement of nuclear power (mostly the ageing capacities) indicates that countries with high share of nuclear capacities may encounter difficulties in exporting their production.

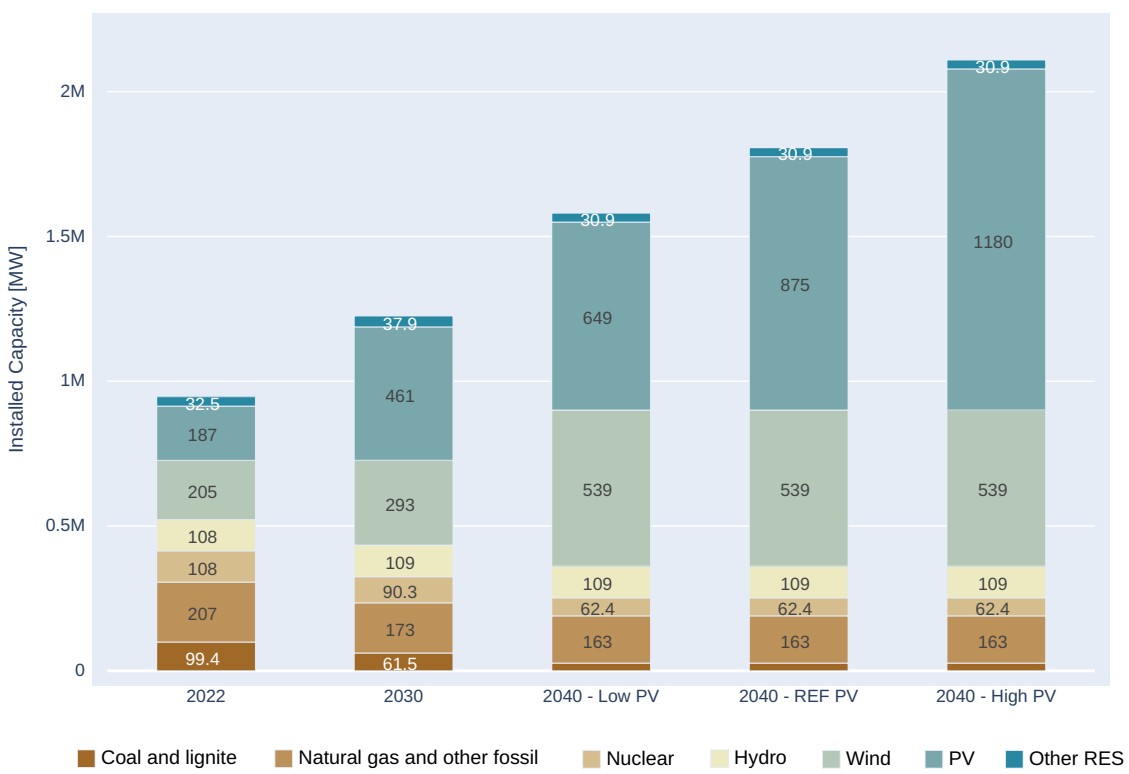

**Fig. 3 | Installed electricity generation capacities in EU27 for 2022, and its evolution with variable share of vertical bifacial PV technology deployment (0 and 50%) by 2030 and 2040.**

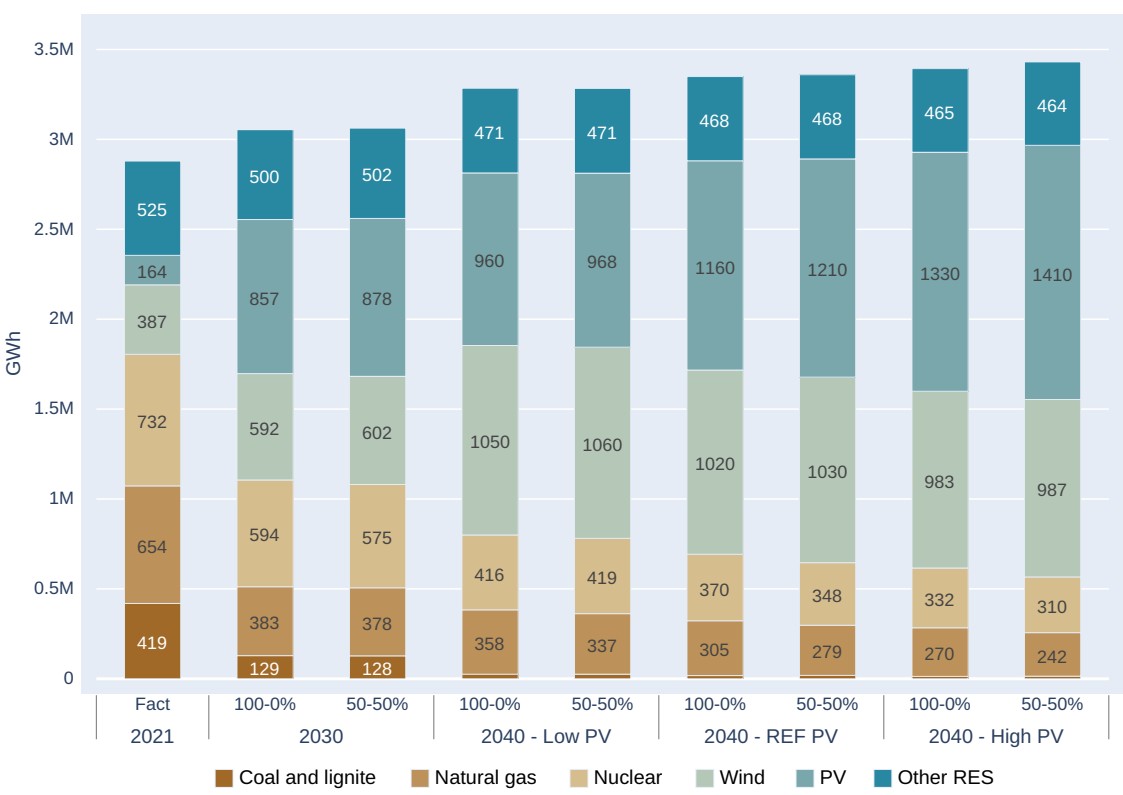

**Fig. 4 | Electricity generation mix and total electricity consumption in EU27 for 2021, and its Evolution with variable share of vertical bifacial PV technology deployment (0 and 50%) by 2030 and 2040.**

The change in production level in the EU is smaller than the change in consumption, due to the increased net export of the EU to outside regions in the bifacial heavy scenarios.

The next sections will elaborate on the power sector impacts on curtailment, wholesale prices, and system costs, which demonstrate even more pronounced impacts than the substitution levels.

## Impact of vertical bifacial PV technology deployment on curtailment

Generation curtailment refers to the reduction in power generation that occurs when there is an excess of electricity on the grid. Curtailment levels, particularly for weather-dependent electricity generation is an important indicator for assessing both the present and future electricity systems. These levels exhibit a strong correlation with the integration cost of RES generators, such as addressing the need for balancing energy, especially in the typical case the need of curtailment occurs after the day-ahead market closure and therefore it requires the activation of capacity/energy in the balancing market. Moreover, they also indicate the negative influence of high PV and wind penetration on the wholesale electricity prices. Nonetheless, it is evident that these adverse effects could be substantially scaled down. In the long term, addressing this issue can be achieved through strategies like increasing the share of demand response mechanisms including heating and transport sector[55], expanding storage capacity, or enhancing hydrogen production capabilities. However, in the short term, the prevalence of this situation is expected to increase, signing for the imperative need for increasing the deployment of flexibility solutions.

Figure 5 compares the energy output of standard PV installation with the various percentages of East-West bifacial PV systems. In principle, typical inclined south faced PV systems generate slightly higher amounts of energy than the East-West bifacial PV. However, the modelling output reveals that the actual PV energy that is fed in the power system increases as the share of East-West bifacial PV system increases that is mainly due to the lower occurrence of curtailments. The figure shows that, for 2040-high PV scenario, by increasing the share of bifacial PV panels from 0% to 50% of the capacity allocation, total curtailments can be reduced from 234 TWh to 131 TWh in the Reference PV, while 562 TWh to 406 TWh in the High PV case. The difference in the first case is equal to Belgium's present power production level, while in the latter case is comparable to Poland's power generation.

The results illustrate that with higher PV penetration, aligning more closely with the recent EU policy commitments, east-west faced vertical PV panels can play a favourable role to achieve a more balanced and more integrated power system in the EU by 2040. They have the potential to reduce curtailment levels, thus reduce the overall balancing costs for the whole power system.

## Effects on baseload price dynamics

Another important aspect to consider is the influence of wholesale electricity prices. It is well-known phenomenon that a high penetration of PV and other weather-dependent RES generation can result in reducing wholesale electricity prices, often referred to as the merit order effect[34]. While it can be advantageous from the perspective of consumers, it can have adverse effects on other producers responsible for reserve capacity and balancing services to the power system. Nonetheless, as the levelized cost of electricity (LCOE) for RES continues to decrease, a new equilibrium should emerge. This equilibrium of the market electricity prices would be determined by the interplay between declining LCOE (energy costs) and raising integration costs, signalling the necessary requirements for balancing power needs and technology solutions. Consequently, it becomes crucial to conduct a carefully analysis of the price impacts.

Figure 4 row 3 further illustrates the development of wholesale electricity prices under various shares of vertical bifacial PV. We can

observe that as the share of vertical PV increases, the weighted average wholesale prices decrease. The weighting factor used is the yearly consumption of each modelled country. Additionally, it is worth noting that the differences in the wholesale price impacts are more pronounced in 2040 than in 2030, due partially to lower system costs, substitution of the more expensive resources and the curtailments.

The PV market values (the average price PV producer could receive in their respective production hours) show a significant deterioration. The market values range between 50% and 19% of the baseload prices in the low and high PV case. The low PV market value is close to previous estimates[34], while the second value (high PV case) is considerably lower since much higher deployment of PV is assumed compared to the estimation[34].

## Resulting $CO_2$ emissions

All these developments are similarly reflected in the reductions of $CO_2$ emission achieved in the scenarios. As the proportion of vertical PV increases, $CO_2$ emissions decrease since higher PV production displaces fossil-based power generation. After the big drop in the $CO_2$ emission we can observe (Fig. 5) relatively modest changes in emissions, primarily because fossil-based generation already accounts for a very low share in the scenarios beyond 2030.

## Total system costs

The total system cost of the future power system is one of the most important system indicators of the model. It inherently encapsulates the measure of total social welfare, as it calculates the producer surplus (including both wholesale market and balancing market components), therefore reflects a more holistic approach beyond merely focusing on the generators profit. Additionally, it considers consumer surpluses, reflecting the prices consumers bear for services in the power sector. This measure also integrates the transmission system revenues, which also partly reflects the network costs and benefits. It is essential to note that the EPMM model does not incorporate distribution system operator-level costs and benefits. Furthermore, the total costs also include the cost related to power generation investments, thereby accounting for the increased costs associated with larger PV panel deployment.

Looking at the development of total cost under the various scenarios and years, it becomes evident (Fig. 5 rows 3 and 4) that the increased share of PV deployment leads to a reduction in the total cost of the power system. This decreasing trend reflects the investment requirements of the current EU decarbonisation policy, where the most significant pressure on the power system occurs during the first decade, marked by the strongest growth in RES developments.

Furthermore, it is noteworthy that the rising proportion of vertically oriented PV deployment results in a decrease in the total cost of the power system: In the 2040 Reference PV scenario, there is a decrease of 3 billion Euros when increasing the vertical module share to 50%. In the 2040 High PV scenario, the decrease is more significant, amounting to 3.8 billion Euros.

Over the longer-term, the European power system becomes more sustainable both economically and environmentally. The future system is characterised by lower $CO_2$ emissions, and reduced reliance on imported fossil fuels, and it will be more sustainable in economic term as well, characterised by reducing total system cost and decreasing wholesale electricity prices.

## Effects of integrating vertical bifacial PV on fossil fuel substitution at country level

When examining at the impacts at higher granularity, at country level, it becomes evident that there are significant variations in how the EU Member States' electricity portfolios are impacted by the varying shares of PV installation modalities. The only consistent change across

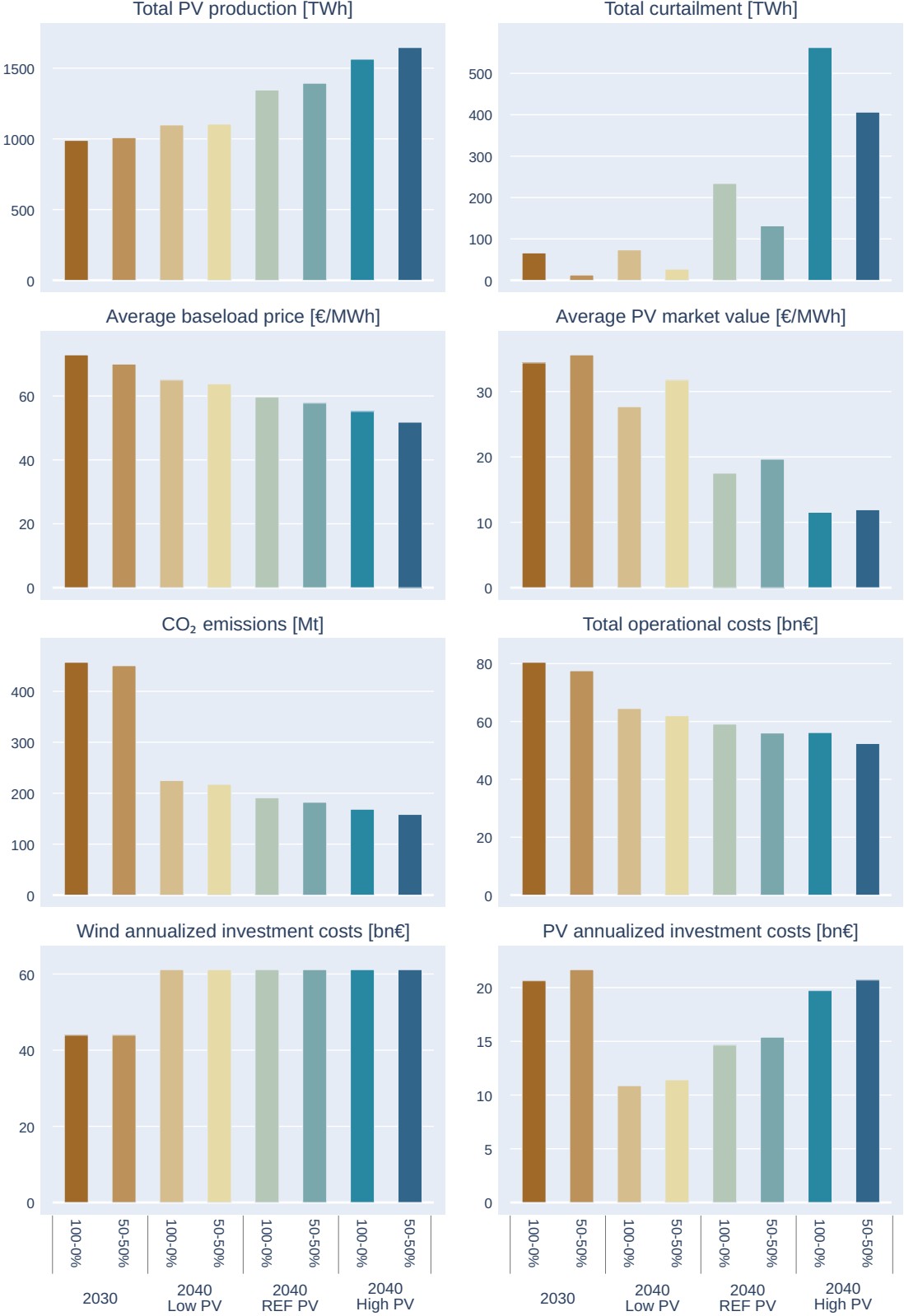

**Fig. 5 | Power sector impacts of varying share of bifacial PV panels**– PV production, generation curtailment, baseload prices, avoided $CO_2$ emissions and total system operational costs, 2040.

all countries is a slight increase in the overall electricity consumption due to the lower prices as indicated by the continuous lines on Fig. 6.

The shares of natural gas in the national portfolios decrease in almost all countries, although to a varying extent, determined the share of natural gas in the generation portfolio of the Member States. One important factor determining the fuel substitution effect is the available net transboundary transfer capacity. In countries where these connection capacities are more available and can further increase their

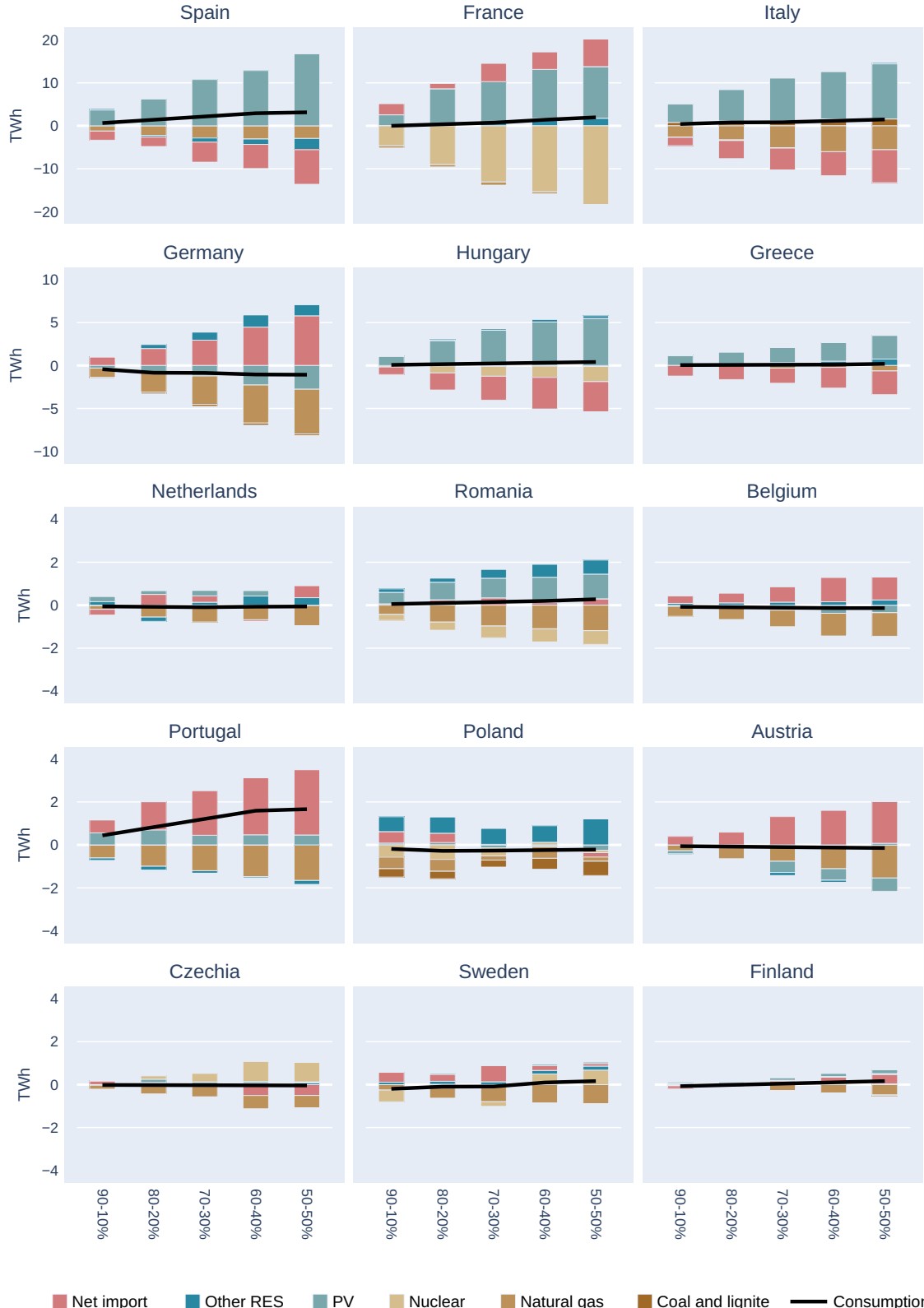

**Fig. 6 | Effects of Integrating Vertical Bifacial PV on electricity mix in various EU countries under the REF and sensitivity PV scenarios in 2040.**

PV shares would increase their export. In contrast, countries with lover PV potential might use import or other renewables. Mediterranean countries experience significant gas substitution levels, with Spain, Italy and Greece due to the increases PV production and export part of this excess. Countries with nuclear capacities generally face reduction of nuclear-based generation, as these capacities are curtailed more, e.g. in France, Romania, Hungary.

In Poland, a higher share of vertical PV results in more technological changes, natural gas and coal-based generation being reduced in the portfolio, while other renewables and import increases.

### Development of electricity prices at country level

In the development of electricity prices, a clear tendency is evident across Europe with the higher share of Vertical bifacial PV: in almost all EU Member States, there is an increase in low-price periods and a decrease in the duration of high-price periods. Simultaneously, the prolonged duration of low-price segments extends across most low-price categories. This extension is a direct result of the increased deployment of Vertical Bifacial PV, which effectively stretches the periods of cheaper PV production.

As a result of these changes, consumer surpluses generally increase, due to that the positive price effects are effectively transferred from producers to consumers. The initial starting price distributions vary significantly among countries. Italy, Greece, Hungary, and Romania are characterised by substantial shares of these high-price segments, while France, Sweden, Finland, and Portugal have lower shares of these high price periods. In most countries, the high-price segments are reflecting the significant gas consumption and/or high import needs. The higher the import rate, the larger the high-price segments in recent years (prior to the crises imports were considerably cheaper). For this group of countries, the installation strategy does potentially decrease their reliance on imports further (as depicted in the effects in Fig. 7) and consequently decrease the price pressure as well.

### Discussion

As the global PV production heads toward 1 TW production capacity, it is paramount to proactively address potential risks to sustain investor confidence and momentum. Many countries are at the verge of encountering counterproductive marks in PV integration, including zero or negative prices, large-scale curtailments, and abrupt regulatory and network operation policies. Until storage capabilities, demand-response and flexibility systems become more widespread, keeping up the momentum of PV installations in novel deployment forms can serve as a bridge to bolster investors' confidence and mitigate market risks.

Mitigating the price drops in spot markets due to more balanced daily massive PV production over extended periods is crucial for maintaining investor confidence and the positive investment environment that has fuelled the PV industry's growth over the past decade. The results show, that favouring vertical bifacial systems reduces peak PV production, and ensues a production profile that covers a larger number of hours, which helps solar-based production maintain higher market value. The deployment of innovative PV concepts like vertical bifacial PV leads to a more balanced production curve compared to the high peak bell-shaped PV generation currently observed in most EU countries. This process already started with the deployment of building integrated and East-West facing PV systems as well as with the agrovoltaics, which could be further scaled up with the bifacial technology. This shift from established approaches also offers easier integration into the EU power system (transmission and distribution), reduces the required grid investments, and enables cross-border trade of solar power for extended periods. These developments can reduce overall system costs and therefore increase societal benefits by avoiding the collective cost of curtailed energy and over-dimensioned grid investments.

The model analysis unveils two prevailing trends in case this disruptive approach is widely adopted: countries with large solar resources are increasing their PV shares, while others (such as Austria, Germany and Belgium) with strong interconnection capacities are boosting imports from neighbouring countries.

Furthermore, this shift has a significant impact on the substitution of gas-coal-nuclear energy in the near future, as demonstrated by the applied model. The displacement of conventional generation extends to a larger number of hours within the day compared to traditional PV system installations, further reducing dependence on imported fuels

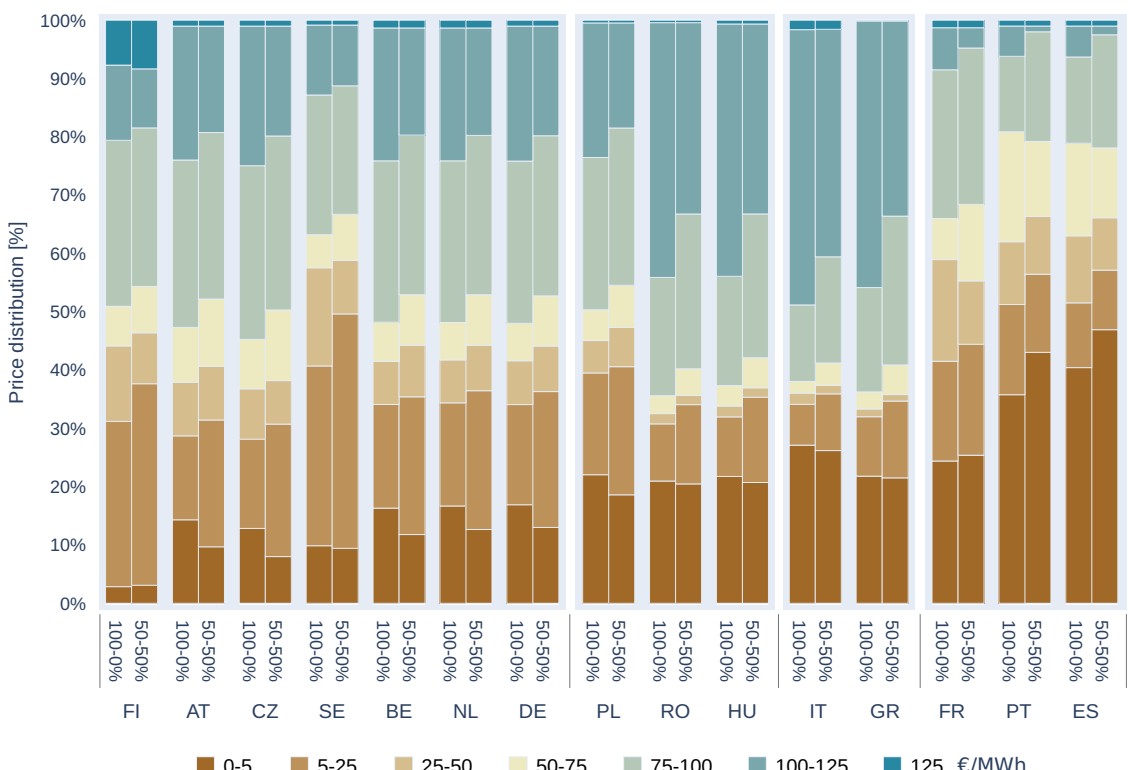

**Fig. 7** | Distribution of Hours in Seven Electricity Price Segments (Ranging from 0 to 0.05 EUR/kWh in Dark Brown, to 1.25 EUR/kWh in Dark Blue) in REF PV- 2040 scenario across 15 European countries.

and GHG emissions. The total costs development underlines that the substantial increase in variable RES capacities is cost efficient due to the reducing fuel and $CO_2$ cost by substituting away fossil-based generation in the European generation mix. The continuing dynamic investment cost reductions of variable RES technologies are key driver for the projected capacity expansion pathway.

If these costs savings are effectively passed on to consumers through enabling regulatory framework and market-based tools such as Contracts for Difference (CfDs), it would enable price reductions across various consumption categories. This would include the extension of low-price periods and –to a lesser extent– a partial reduction in high-price periods due to peakshaving. By largely substituting import-based production, mainly gas based generation at the end of the modelling period, the EU can achieve a more self-reliant and independent electricity portfolio.

The new forms of diverse deployments offer opportunities not only to increase employment in the already robust installer segment but also to create a market niche for PV module industries (bifacial modules, tracking systems, vertical mounting systems on roads and railways). Additionally, it can generate income for farmers, especially if vertical PV installations could obtain easier permission and connection terms for agrivoltaics.

The scalability of such diversified PV electricity has huge implications for hydrogen production (including transport), for other energy reserves, and for the performance potential for heat pumps and wind – PV complementarities. While hydrogen valleys are not analysed in this article, they hold promise for further research. An additional area for future analysis is the inclusion of battery storage within the proposed concepts. As batteries' cost reduces and the market maturity of the technology increases, PV system setups could be coupled with behind-the-meter storage. The merits of such hybrid systems would be further extended as their production could be extended earlier/later during the day and minimised in mid-day in way that compensates for the peaking production of traditional solar PV systems. Equally importantly, hybrid systems would have the potential to provide ancillary services and actively participate in the balancing markets, creating an additional revenue stream for PV.

The findings also have important implications on the performance of the US Inflation Reduction Act[56] and the EU flagship industrial initiatives, namely the European Commission's permitting package, EU Solar PV Industry Alliance, the EU large-scale skills partnership, and the EU Solar Rooftops Initiative. The proposed permission process, which targets a maximum duration of 3 months, can potentially be further reduced for lower-impact installations like vertical PVs. If this market segment develops rapidly, it can create opportunities for a new European PV manufacturing industry, presenting a sizeable market niche and driving the need for improved skill education.

## Methods

### General model description

The EPMM is a 168-hour unit commitment and economic dispatch model covering the electricity systems of 41 European countries shown in Fig. 8. It simultaneously determines the equilibrium values of the wholesale electricity and reserve markets for each hour and market, taking into account the projected weather-dependent renewable generation, the electricity demand, the reserve requirements for each countries, and the technological constraints and costs (minimum operating and off-time, minimum/maximum load level, start-up / shutdown costs, variable costs of generation) for electricity generation and transmission. Using these inputs, the model predicts the operating status of the power plants every hour of the week (covering almost 3,500 power plants), the volume of generation at the operating units, the amount of capacities set aside for upward-regulation and downward-regulation, the operation of reservoir hydropower plants, the flows on all cross-border interconnectors, and the wholesale market price of electricity and the price of the upward and downward reserve capacities in each country.

There are 41 countries modelled in EPMM: in these countries (indicated with an orange background in Fig. 8) prices are derived from the demand-supply balance, while on outside markets (indicated with yellow background) we assume exogenous prices.

There are three types of market participants in the model: producers, consumers, and traders. All of them behave in a price-taking manner: they take the prevailing market price as given and assume that their actions have a negligible effect on this price.

The EPMM models 3500 power plant units operated with 12 different fuels: natural gas, coal, lignite, heavy fuel oil (HFO), light fuel oil (LFO), nuclear, biomass, geothermal, hydro, wind, solar, and tide and wave. Each plant has a specific marginal cost of production, which is constant at the unit level. In addition, generation capacity is constrained at the level of available capacity. Renewable technologies are modelled in an aggregated way.

Power flow is ensured by 110 interconnectors between the countries, where each country is treated as a single node, thus no domestic power system constraint is taken into account. NTC values are used to indicate trading possibilities, seasonal differences are included in the modelling based on historical data from ENTSO-E Transparency Platform. Future investments are based on data from ENTSO-E's latest Ten-Year Network Development Plan (TYNDP). NTC values, cost characterisation of vRES technologies (e.g. investment costs, fix costs, annualised costs), emission factors are presented in the Supplementary Data 1–5.

Consumers are represented in the model in an aggregated way: by different price-sensitive demand curves for each modelled market. The inverse relationship between prices and the quantity consumed is approximated by a downward sloping linear function. Traders connect the production and consumption sides of a market, through exporting electricity to more expensive countries from cheaper ones.

Taking into account the short-term marginal cost for all available power plant units merit order curves are calculated for each market. With the demand curve and the constraints on international trade all input parameters are set. After this process, the model maximises total welfare of the whole assessed region. The model provides the equilibrium (wholesale) electricity prices for each market, the trade on each interconnector and the production of each power plant unit as output as shown in Fig. 9.

### Supply side of the model

As perfect competition is assumed when the supply curve is formed in the model all units provide their production on a marginal cost basis. To calculate marginal costs unit specific $CO_2$ emission cost, energy tax (if any), fuel cost and variable OPEX are summed up.

For all given technologies (e.g. OCGT, CCGT, thermal) commissioning date defines the efficiency, the self-consumption and the variable OPEX cost for all units. Using the fuel prices as an input total fuel costs are calculated taking into account the above parameters. $CO_2$ costs are based on the calculated emission level and the $CO_2$ quota prices, and all these costs are then added to the total energy tax paid and the variable OPEX.

It is important to note that only short-term marginal costs are considered, the model does not analyse whether long-term operation is profitable or not. It is possible, that some units remain operational even if they provide electricity in a few hours per year.

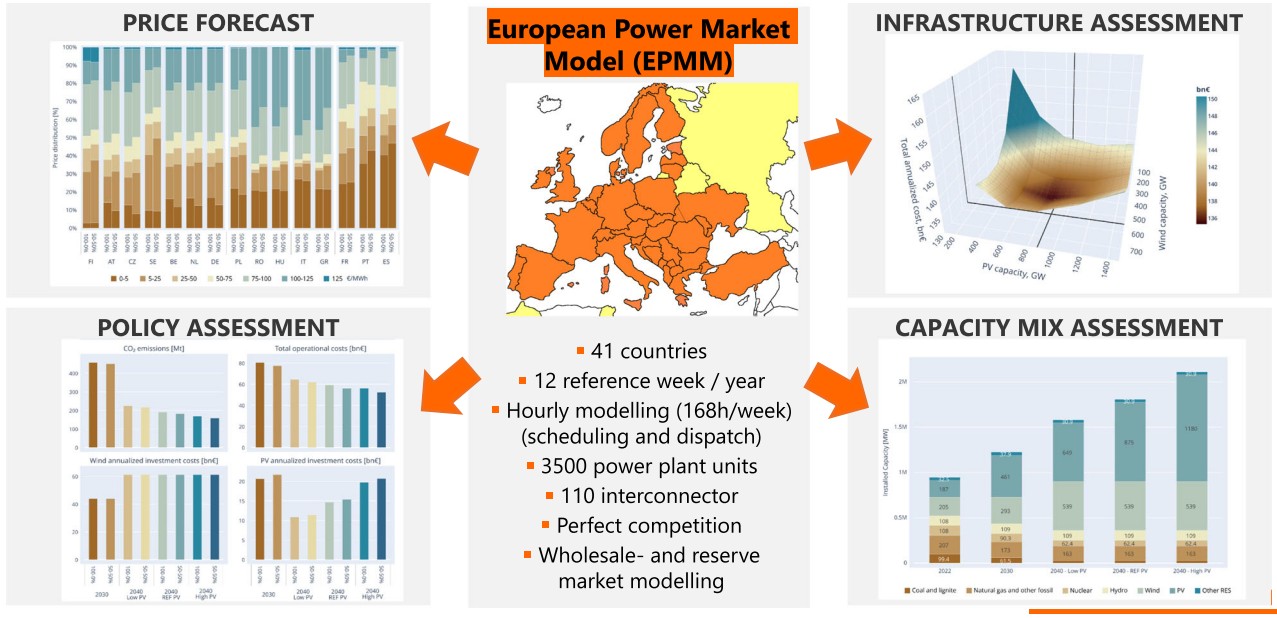

**Fig. 8 | EPMM model flow chart.**

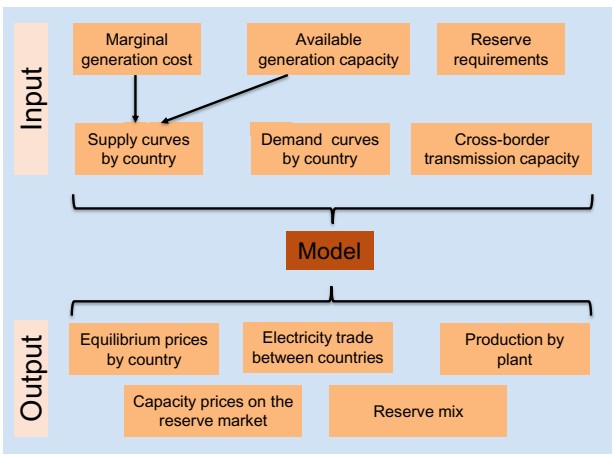

**Fig. 9 | Functional framework of the EPMM model.**

Power plant units are available until the end of their (pre-defined) lifetimes.

In the case of renewable power plants (biomass, geothermal, hydro, wind, solar, tidal), we calculate zero marginal cost on the product market, production depends only on availability. This is estimated based on historical production data from previous years (2006-2017). Renewable production (at a predetermined cost) can be curtailed. In the case of participation in the reserve market, we assume linear growth for wind and solar, with wind plants participating in the downward regulation up to 25% of their production and solar plants up to 15% of their production by the mid-2030s. The situation for hydropower is specific. With the exception of run-of-river plants, all hydropower plants (reservoir and pumped storage) optimise their production and reserve market participation for a full week, taking into account as a limit the maximum available production expected based on historical production data from previous years. These are used to determine the reserve market prices, while the marginal cost of the product market is zero on the supply curve.

A regression model is used to forecast the future level of required reserve capacities: based on 5 years (2017-2022) and 16 countries of data (AT, BE, CH, CZ, DE, DK, ES, FR, HR, HU, NL, PL, PT, RO, RS, SI, SK), upward and downward reserve requirements are forecast for each modelled country and each modelled hour, as a function of system load and weather-dependent renewable generation capacity.

### Demand side of the model
Demand per country is an exogenous input to the model. Historical data are used to assume a specific demand pattern for each modelled country over the year, and assumptions are made about the actual level of demand based on forecasts from international institutions, literature and strategy documents. Between years, the initial demand path is unchanged, but it is endogenously shaped by (pumped) storage and demand side management (DSM) in each modelled year. For DSM, based on literature, it is assumed that by 2050, 25% of the average hourly consumption in a given year can be reallocated to other hours, and 10-10% of this amount can provide for upward and downward reserve capacity services. Between 2020 and 2050 we assume linear growth.

### Equilibrium
Based on the capacity and marginal cost values of the generating units, and considering which units are in reserve, the supply function is determined for each modelled hour in each country. Demand can be met from neighbouring countries in addition to domestic supply, up to the limit of cross-border capacity. In the model, each market (country) is represented by a node, and the cross-border capacities are represented by estimated NTC (net transfer capacity).

### Solar irradiation and PV production input data
The PV production data for the different modules were downloaded from the Joint Research Centre EC PVGIS tool[57].

Average monthly sum of global irradiation per square meter received by the modules of the given system [kWh/m2/mo] data for optimised and for East West facing vertical & bifacial PV modules (Fig. 10).

The data are presented in a database format in Supplementary Data 1–5.

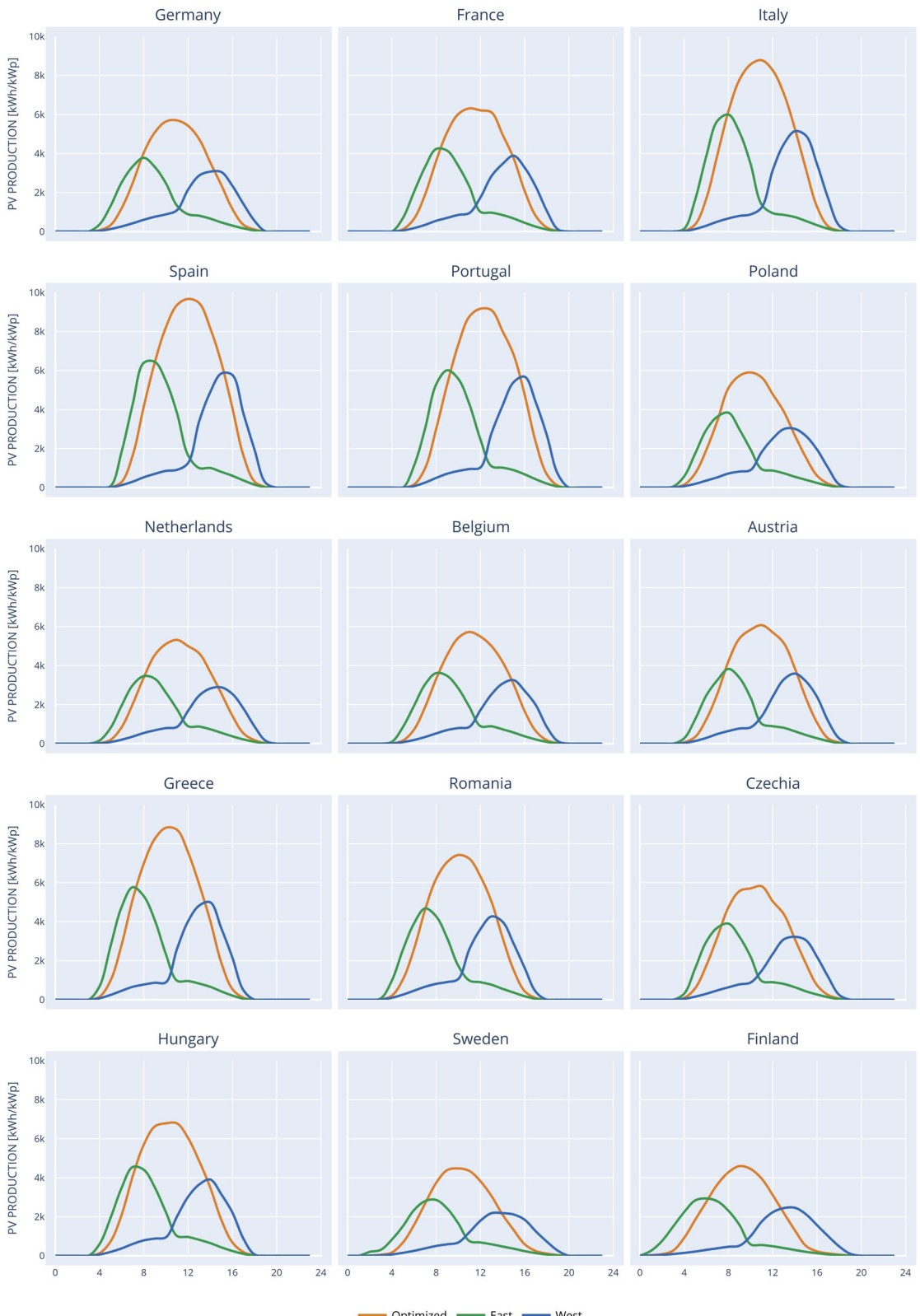

**Fig. 10 | The change of PV output profile from South facing (orange lines) to East&West facing (green&blue lines) PV mudules.** The photovoltaics geoinformation system (PVGIS) generated PV output profiles samples for a representative set of 15 countries that was used in the modelling inputs (with other countries and seasons).

## Data availability

Only openly available data were used in this study. These and any model equations used to produce the necessary data and figures in this study that can reproduce the results in any modelling languages are made available via Zenodo under accession link https://doi.org/10.5281/zenodo.12530393. Input data used throughout the study are

provided in Supplementary Data 1-5. Output data of the applied model displayed in Figs. 2–7 are provided in Supplementary Data 6.

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

## Acknowledgements

We thank Tony Sample (Joint Research Centre (JRC)—European Commission), for insightful comments on East West facing bifacial module characteristics used in this article. M.M.G and F.F. are consultants for JRC—European Commission. The views expressed are purely those of the authors and may not in any circumstances be regarded as stating an official position of the European Commission. No funding was used in this research.

## Author contributions

L.S. and S.S. Conceptualisation, Methodology, Writing manuscript, Data interpretation. M.M.G. and A.J.W. Writing manuscript, Measurement data validation. I.K.: Writing manuscript, Market data validation. A.M.: Data collection and analysis, Data visualisation, Modelling. F.F.: Data collection, Visualisation. All authors have reviewed and approved the final version.

## Competing interests

The authors declare no competing interests.
