## [Peer Review File · Nature Communications]

Impact of large-scale deployment of vertical bifacial photovoltaic on European electricity market dynamicsREVIEWER COMMENTS

Reviewer #1 (Remarks to the Author):

The paper deals with a subject that falls within the scope of the journal. Please avoid bulk referencing, provide a brief discussion for all referenced work. As depending on the type of installed system (small, medium, residential, commercial) different options are available for financing and contractual reasons, a brief categorization is requested. In the "reshaping of the energy landscape" please provide a brief discussion on relevant work available in the literature that deals with PV systems aiming for maximum generation. Also please provide citations for the various statements done. In the alternative deployment options sections the authors should provide a more concise overview of what is the main aim of the paper and its novelty. Figure 1 should be cut into more as in its current size it is not easily readable. In table 1 please provide current prices as well for comparison. It is not clear if in the simulation effects of varying vertical bifacial PV technology Deployment the shading from adjacent buildings or structures has been taken into consideration. Furthermore it is not clear if the same parameters have been considered for all EU27 countries or not and why. In Figure 4 the emission factor used for estimating the CO₂ should be provided. In general a table with the values used for all the calculations presented in figure 4 should be provided in the text along with their references. Figure 5 is not clear, please use bigger fonts and/or break it in more figures. In section 1.2 please provide the 5 years and the 16 countries used to forecast the future level of reserve capacities.

Reviewer #2 (Remarks to the Author):

Key results

The paper uses an economic dispatch model to compare scenarios of saturation of the installed PV capacity in Europe with non-standard deployments like vertical bifacial installations. The authors find a decrease in curtailment and wholesale prices, and a slight decrease in emissions and operating costs expenditures.

Validity

The study certainly provides interesting insights into the potential of vertical bifacial PV installations. Yet, the claim that this would be a short-term solution compared to flexibility in the system is in my opinion a bit of a bold statement. Already for 2030, the European Commission foresees large quantities of green hydrogen production from renewable electricity. 2040 cannot be called short-term anymore since the EU targets climate neutrality already in 2045. Also, it is unclear if and when the tested technology will be available at scale in comparison to storage technologies, grid expansion, and demand-side flexibility from electrolysis, heat pumps, and electrified transportation. Is it realistic to scale up vertical PV to the extent tested, given the problems with building codes, availability of suitable areas, etc.?

The introduction mentions some alternative scenarios but fails to explain them in some detail at least. Given the recent announcements for offshore wind development, this should also be considered as an alternative. Other works show the significant impact, flexibility from new demand from electrification can have on the integration of fluctuating renewables (cf. e.g., <https://doi.org/10.1016/j.energy.2023.127832>).

Significance

The study certainly contributes to the literature, yet, in its current state, I think it is a bit too simple. The introduction mentions alternative deployments with the title claiming "disruptive deployments". The simulation itself only seems to contain east/west-facing bifacial vertical PV and

disregards other mentioned options like tracking. The order of magnitude of changes in the reported parameters also seems underwhelming and not living up to the expectations of the title.

Also, I am missing a more detailed naming of alternatives like flexibility (see Section Validity) and their impact on market values in a sector-integrated system (cf. e.g., <https://doi.org/10.1016/j.apenergy.2020.115985>) but also the discussion on two-sided CfDs (cf. e.g., <https://doi.org/10.1038/s41560-023-01401-w>).

Data and methodology

Why has only a dispatch but no investment model been used? The authors mention this aspect but the only reason stated is the necessary temporal resolution. An investment model could also be coupled with a dispatch model. This way not only the static effect of different saturations of alternative PV installations could be shown but also the interaction with the larger system and a better message about the significance compared to other solutions could be made.

How were the PV targets for 2040 derived? The NECPs are mentioned for 2030 but it is unclear how the levels for 2040 were calculated, especially concerning the country-specific values.

What are the cost assumptions for the considered PV technology? I could not find this in the manuscript – only a note that CAPEX for new installations is considered in the model. How is this delimited in the scenarios? This did not become clear to me.

Please explain your scenarios in more detail. Are the stated saturation rates in relation to all installed capacity or only to newly added capacity? What shares do east- and west-facing installations have? Why were no other technologies like tracking considered?

The price of coal in Table 1 is stated in USD. For consistency, this should also be given in EUR (converted).

How were the NTCs for 2040 derived? The TYNDP is mentioned, yet, it does not lead until 2040 currently. Were there any additional expansions considered?

Analytical approach

In general, I find the approach too static, only using a dispatch and no investment model and not showing any counterfactuals for other integration options.

In the results section, I am missing information on the market value of renewables, specifically PV. Ideally, this should be distinguished by "optimized" and "bifacial vertical". The introduction mentions the cannibalization of renewables but fails to show this effect in the analysis.

Suggested improvements

As mentioned, the analysis is too static and would gain significantly from testing counterfactuals. See also my comments on data and methodology.

The abstract should be improved. Currently, it does not become clear what the contribution of the paper is and what novel/diversified installations have been considered.

I am also strongly arguing in favor of publishing the methodology code under an open-source license and (or at minimum) the resulting data for further use in energy system modeling studies.

The quality of the graphs needs to be improved. Most of them are illegible and should be provided as vector graphics or at least with a significantly higher resolution.

Clarity and context

The paper is written in a very clear and accessible way. It is well structured and easy to follow, and

the line of arguments is well presented. I do recommend a thorough proofread to eliminate the few existing typos.

References

While the paper references previous literature to the extent serving its results, I am missing some aspects: (1) alternative instruments like two-sided CfDs (cf. e.g., <https://doi.org/10.1038/s41560-023-01401-w>), (2) alternative integration options like flexibility from electrification (cf. e.g., <https://doi.org/10.1016/j.energy.2023.127832>), and (3) the effect of system integration on market values (cf. e.g., <https://doi.org/10.1016/j.apenergy.2020.115985>).

Reviewer #3 (Remarks to the Author):

Comments to the Author

This paper proposed "Disruptive deployment of PV technology unlocking the full potential of solar energy in the merit order". The presentation of the submitted manuscript and the proposed approach requires major improvements of the paper. This reviewer would like to see the effort of the authors to revise the paper and addressing the following questions:

- 1) Clarity is required in the abstract on the contribution of the proposed work.
- 2) The introduction needs to be revisited. It is quite elaborate and lacks direction with a clear research gap. There is repetition of information at many places. The contribution of the work must be thoroughly rewritten with more clarity.
- 3) What is the major contribution of the paper? More and clear presentation of the paper contribution in abstract and introduction is necessary.
- 4) The novelty of the work must be clearly addressed and discussed, compare your research with existing research findings and highlight novelty, (compare your work with existing research findings and highlight novelty).
- 5) Why did the authors focus the paper on disruptive deployment of PV technology?
- 6) The methodology is not very clear, the author needs to clarify and justified.
- 7) The supply and demand side model is not clear, authors need to justified.
- 8) In case studies, some assumptions need to be clarified and justified.
- 9) The result portion needs to be re-written from scratch, to bring about flow of thought.
- 10) More updated references for the years 2022 and 2023 can be included for literature review.
- 11) Conclusion section is missing some perspective related to the future research work,
- 12) Please improve the quality of figures
- 13) English language should be carefully checked and carefully check paper for language typos.

Dear Editor, Dear reviewers,

The authors would like to thank the constructive comments and recommendations provided on the manuscript. By addressing and reinforcing the analytical section and partially redirecting the discussion based on the feedback of the reviewers, we have enhanced the communication of the results and highlighted the multidisciplinary approach of the engineering, commercial and techno-economic novelty to the different stakeholder communities. Please find attached the detailed answers to each recommendation.

Reviewer #1 (Remarks to the Author):

We would like to thank reviewer 1 for the constructive critique, which has added value to our research and strengthened the core message of our paper provided. The feedback have significantly contributed to the evolution of our study, which examines the transformative period that the power systems, energy markets, and policy landscapes are currently experiencing. This transformation is primarily driven by the urgent need to mitigate climate impacts from fossil fuel-based emissions and the pivotal role of these systems in spearheading decarbonization through the adoption of renewable energy sources.

The paper deals with a subject that falls within the scope of the journal.

Please avoid bulk referencing, provide a brief discussion for all referenced work.

Answer: We have split up the bulk references and we have provided separate discussion for each reference as suggested by the reviewer.

As depending on the type of installed system (small, medium, residential, commercial) different options are available for financing and contractual reasons, a brief categorization is requested.

Answer: The authors have acknowledged the need for different financing and contractual options based on the type of installed system (small, medium, residential, commercial). They have provided a brief categorization to accommodate these diverse needs and considerations.

In the “reshaping of the energy landscape” please provide a brief discussion on relevant work available in the literature that deals with PV systems aiming for maximum generation. Also please provide citations for the various statements.

Answer: We have restructured the discussion in the "reshaping of the energy landscape" subsection to provide a brief overview of relevant literature on PV systems aiming for maximum generation. Additionally, we have included citations to support various statements made in the discussion. Our revisions also address the consequences of the concentration of production around midday, highlighting system integration, technical, and market challenges. Furthermore, we have expanded our references to encompass regions beyond the EU.

- Klaaßen, L. & Steffen, B. Meta-analysis on necessary investment shifts to reach net zero pathways in Europe. *Nat. Clim. Chang.* **13**, 58–66 (2023)

- McKinsey&Company. *Renewable-energy development in a net-zero world*. <https://www.mckinsey.com/industries/electric-power-and-natural-gas/our-insights/renewable-energy-development-in-a-net-zero-world> (2022).
- Prokhorov, O. & Dreisbach, D. The impact of renewables on the incidents of negative prices in the energy spot markets. *Energy Policy* **167**, 113073 (2022).
- Aust, B. & Horsch, A. Negative market prices on power exchanges: evidence and policy implications from Germany. *Electr. J.* **33**, 106716 (2020).
- McKinsey&Company. *Renewable-energy development in a net-zero world*. <https://www.mckinsey.com/industries/electric-power-and-natural-gas/our-insights/renewable-energy-development-in-a-net-zero-world> (2022).
- Commission, E. *Fit for 55*. <https://www.consilium.europa.eu/en/policies/green-deal/fit-for-55-the-eu-plan-for-a-green-transition/> (2022).
- Solar Power Europe. *It is not curtailment. It is waste*. https://api.solarpowereurope.org/uploads/FINAL_Solar_Power_Europe_Solar_Energy_Waste_Letter_August_2023_d9b2c2d2aa.pdf?updated_at=2023-08-02T07:22:20.469Z (2023).
- Nicolai, E.; Menegatti, S. *New challenges for DSOs in the energy transition*. EUI Florence School of Regulation. <https://fsr.eui.eu/new-challenges-for-dsos-in-the-energy-transition-more-grid-investments-increased-network-planning-and-forecasting-of-new-technologies/> (2023).
- Peña, J. I., Rodríguez, R. & Mayoral, S. Cannibalization, depredation, and market remuneration of power plants. *Energy Policy* **167**, 113086 (2022)

In the alternative deployment options section the authors should provide a more concise overview of what is the main aim of the paper and its novelty.

Answer: We extensively rewrote the deployment option section focusing on the aim and novelty. We have inserted the following:

“In the shorter time horizon, it is crucial to identify quick, no-regret options capable of accommodating the unparalleled scale-up penetration of PV technology. The present paper explores such a no-regret option: the potential benefits of installing large quantities of PV systems in a more diversified and sustainable manner. This strategy capitalizes on several factors, including the adoption of new module technologies like bifacial modules, and innovative installation practices such as non-standard orientations, vertical PV. “

...

“From system integration point of view vertical East-West facing PV could be considered as bigger transformation from South facing PV than the conversion between the off-shore and on-shore wind: it produces 30% in the 3 midday hours in contrast to the South facing PV with near 70%

...

“Vertical installation also addresses the sensitive issue of limited land availability for PV deployments, enabling the utilization of a broader range of areas, including applications in agricultural land and greenhouses, also known as agrivoltaics. Additionally, these systems can be integrated into linear infrastructures like highways and incorporated into building structures as building-integrated PV (BIPV) solutions.”

“To further advance the development of PV systems at even higher shares, it requires solutions that can effectively manage solar output variations and alleviate the growing challenges associated with mounting balancing and integration. While storage-based solutions are expected to play a significant role in the long term, this paper highlights the opportunities presented by emerging PV technologies and innovative system designs to minimise the overall costs of system transformation.”

“This paper presents a model-based approach aimed at illustrating the diverse impacts of deploying high shares of novel and diversified solar PV installation.”

Figure 1 should be cut into more as in its current size it is not easily readable.

Answer: We have simplified the figure removing redundant information and enhancing visibility.

In table 1 please provide current prices as well for comparison.

Answer: current prices for the year 2023 are provided for comparability.

It is not clear if in the simulation effects of varying vertical bifacial PV technology Deployment the shading from adjacent buildings or structures has been taken into consideration. Furthermore, it is not clear if the same parameters have been considered for all EU27 countries or not and why.

Answer: We would like to clarify that the simulation incorporates the PV outputs dataset from the PVGIS Information System as its primary input. This dataset comprehensively includes the effects of shading due to geographical features such as terrain contours, mountains, and vegetation. Consequently, these forms of shading have been duly integrated into our simulation results.

However, it is important to note that the impact of shading from buildings has not been specifically factored into the simulation. We have determined that this omission is justified because the predominant focus of our analysis is on rural areas. In these locales, the presence of buildings is significantly less dense, and thus the potential shading they might cause is considered to have a minimal or negligible effect on the simulation outcomes.

Regarding the application of parameters, we acknowledge that there is a degree of variation that reflects the distinct conditions in each of the EU27 countries. Parameters such as solar irradiation, investment considerations, and dispatch model specifications do indeed differ from one country to another to accurately represent the diverse environments and economic landscapes across the European Union. These country-specific parameters have been documented and are readily accessible in the supplementary material accompanying our study.

We hope this response addresses your concerns and provides a clearer understanding of the methodologies employed in our simulation.

In Figure 4 the emission factor used for estimating the CO2 should be provided. In general, a table with the values used for all the calculations presented in figure 4 should be provided in the text along with their references.

Answer: the technology specific emission factors are provided in the support online material.

Figure 5 is not clear, please use bigger fonts and/or break it in more figures.

Answer: We have made the necessary revisions to enhance the clarity of the figure. We have increased the font size to improve legibility and ensure that all text within the figure is easily readable. Additionally, we have re-evaluated the layout and presentation of the data to ensure that the figure communicates the information effectively. We trust that these adjustments address your concerns and that Figure 5 now meets the standards for clarity and readability.

In section 1.2 please provide the 5 years and the 16 countries used to forecast the future level of reserve capacities.

Answer: We have updated the section to include the specified information. The countries analysed in the forecast are Austria (AT), Belgium (BE), Switzerland (CH), Czech Republic (CZ), Germany (DE), Denmark (DK), Spain (ES), France (FR), Croatia (HR), Hungary (HU), Netherlands (NL), Poland (PL), Portugal (PT), Romania (RO), Serbia (RS), Slovenia (SI), and Slovakia (SK). The time period used for the forecast spans from 2017 to 2022.

This elaboration has now been incorporated into the text to provide a clearer understanding of the scope and basis of our forecasting model.

Reviewer #2 (Remarks to the Author):

We would like to thank Reviewer #2 to point out the importance of incorporating an investment model into our analysis. By doing so, we have significantly enhance the robustness of our study, more accurately reflecting the dynamic nature of capacity, cost, and changes in technology values.

Key results

The paper uses an economic dispatch model to compare scenarios of saturation of the installed PV capacity in Europe with non-standard deployments like vertical bifacial installations. The authors find a decrease in curtailment and wholesale prices, and a slight decrease in emissions and operating costs expenditures.

Validity

The study certainly provides interesting insights into the potential of vertical bifacial PV installations. Yet, the claim that this would be a short-term solution compared to flexibility in the system is in my opinion a bit of a bold statement.

Already for 2030, the European Commission foresees large quantities of green hydrogen production from renewable electricity. 2040 cannot be called short-term anymore since the EU targets climate neutrality already in 2045. Also, it is unclear if and when the tested technology will be available at

scale in comparison to storage technologies, grid expansion, and demand-side flexibility from electrolysis, heat pumps, and electrified transportation. Is it realistic to scale up vertical PV to the extent tested, given the problems with building codes, availability of suitable areas, etc.?

Answer: We acknowledge the ambitious projections by the European Commission for green hydrogen production by 2030 and the EU's target for climate neutrality by 2045. Our use of "short-term" was intended to reflect the immediate and near-term time frame, leading up to 2030. We agree that the term "short-term" could imply a narrower time scope, and we appreciate your guidance on this matter.

Our study aims to provide an overview on the situation on the energy market development. While solar and hydro based electricity production installations grow in an unprecedented pace, the foreseen flexibility and storage capacities are lagging behind, due to the infrastructure development and other needs.

The model results we present show that the plans of tripling the PV in the next 7 years (which are currently on track) would benefit by changing part of the vast PV deployment to East-West facing bifacial/vertical (in roadside, agrivoltaics), at least in the in the short term. In the long term, storage and sector coupling and the associated infrastructures (electrolysers, hydrogen network developments) may gradually take over the decarbonization efforts. The investment model that was integrated to the dispatch model also supports that this change would mitigate the decreasing PV economic value.

In the text we show studies underlying renewable technologies are ready to scale up to multi-TW levels, and we show solutions delivered at hundreds MW size based on bifacial PV modules. We see the fastest scaling up potential is with the agrivoltaics and roadside PV module, as they are quite similar to freestanding PV installations, but having lower land-use impacts and provide additional income for farmers and connections to e-mobility.

In summary, while we recognize the long-term importance of storage, grid expansion, and demand-side flexibility, our research suggests that vertical bifacial PV technology can play a crucial role in the transition phase towards a fully decarbonized energy system. We have adjusted the text to more clearly articulate these points and to address your concerns about the timeframe and scalability of the technology.

The introduction mentions some alternative scenarios but fails to explain them in some detail at least. Given the recent announcements for offshore wind development, this should also be considered as an alternative.

Answer: In the new, revised version both PV and wind development is based on the investment module decision and not on the NECPs (National energy and climate plans). Especially for PV, a min-max range is determined to show alternative pathways. We focused the study on the market integration problem that is caused by the daily production pattern of solar PV, and we show the shares of optimal PV-wind development range in the overall portfolio.

Other works show the significant impact, flexibility from new demand from electrification can have on the integration of fluctuating renewables (cf. e.g., <https://doi.org/10.1016/j.energy.2023.127832>).

Answer: In the revised version several studies have been introduced and discussed on the flexibility demand and variable renewables interactions.

Significance

The study certainly contributes to the literature, yet, in its current state, I think it is a bit too simple. The introduction mentions alternative deployments with the title claiming “disruptive deployments”. The simulation itself only seems to contain east/west-facing bifacial vertical PV and disregards other mentioned options like tracking. The order of magnitude of changes in the reported parameters also seems underwhelming and not living up to the expectations of the title.

Answer: At the TWh scale level of the projected solar PV deployment, the effects of different shares of the typical and the new way of deployment are given between the ranges of 100%-0% to 50%-50%. The important parameter for the optimisation process is to what extent the midday peak of the typical PV production can be distributed outside the midday period (flattening the bell-shaped production curve), the internal allocation of we reach the 10-20-30-40-50% projected shares by the potential alternative applications of one or two axis tracking and East-West vertical bifacial does not influence the cost optimality as they are characterized by very similar costs.

Also, I am missing a more detailed naming of alternatives like flexibility (see Section Validity) and their impact on market values in a sector-integrated system (cf. e.g., <https://doi.org/10.1016/j.apenergy.2020.115985>) but also the discussion on two-sided CfDs (cf. e.g., <https://doi.org/10.1038/s41560-023-01401-w>).

Answer: The various flexibility mechanisms and the two-sided CfDs have been introduced in the paper and their potential are discussed now extensively and the impacts of the market developments on market value of PV has been calculated in the integrated model.

Data and methodology

Why has only a dispatch but no investment model been used? The authors mention this aspect, but the only reason stated is the necessary temporal resolution. An investment model could also be coupled with a dispatch model. This way not only the static effect of different saturations of alternative PV installations could be shown but also the interaction with the larger system and a better message about the significance compared to other solutions could be made.

Answer: Thank you very much for the suggestion. In the revised version the investment module is used to plan capacity expansion for the modelled period, and its functioning is based on cost minimisation over the planning time period. This approach has the advantage of planning the overall RES expansion in a cost optimal way, in contrast to the NECP based approach, where capacity uptake is a more of a political decision. The new results suggest that PV capacity expansion optimum is around 875 GW of installed capacity of PV in 2040, similar to previous figures. However, there is a

substantial increase in wind generation to 539 GW level at the optimum, some 110 GW higher than would be based on the NECP trajectory. This shows that wind generation would have a higher role, than the NECP values would suggest. The other message of the investment-based approach is that system cost is quite flat over a wide range of installed PV capacities, so we selected a range of PV capacities from a minimum of 649 GW to a maximum of 1176GW capacities, that divert annualised system cost in a +/- 2% range, to reflect on the uncertainties of the investment decisions. This range of estimate enables us to make the assessment less deterministic and static. Consequently, all result figures are updated given the range for a minimum and maximum capacity estimate and helps to draw more robust conclusions.

See Section on Modelling Framework for Baseline Scenarios and Figure 3-7.

How were the PV targets for 2040 derived? The NECPs are mentioned for 2030 but it is unclear how the levels for 2040 were calculated, especially concerning the country-specific values.

Answer: in the revised version the investment module determines the cost optimal investment both in PV and wind generation, and it is not based on the NECP targets. This is now explained in the Modelling framework section of the revised paper.

What are the cost assumptions for the considered PV technology? I could not find this in the manuscript – only a note that CAPEX for new installations is considered in the model. How is this delimited in the scenarios? This did not become clear to me.

Answer: We include cost estimates an online annex (see Technical parameters part) with all the output results data, and we also include the cost assumption on PV and wind technologies. The decreasing trends in investment cost is an important input to the cost optimisation (in the investment module) determining the cost optimal level of RES technologies in the future capacity mix.

Please explain your scenarios in more detail. Are the stated saturation rates in relation to all installed capacity or only to newly added capacity? What shares do east- and west-facing installations have? Why were no other technologies like tracking considered?

Answer: In the reference scenario the Variable Renewable Energy (vRES) capacities and the generation of electricity by technology are based on the optimisation of the model, which minimises the total welfare of the society (consumers, generators). In comparison to that, various levels of bifacial modules are assumed up to 50% of the new investments in PV and the results for these sensitivity cases are introduced in the Results section. This is explained more clearly in the Modelling framework and Simulation effects sections.

The price of coal in Table 1 is stated in USD. For consistency, this should also be given in EUR (converted).

Answer: the tables updated with EUR figure, also 2023 actual prices are provided.

How were the NTCs for 2040 derived? The TYNDP is mentioned, yet, it does not lead until 2040 currently. Were there any additional expansions considered?

Answer: The MSs TYNDPs generally look forward up to 2035 now, therefore we used the planned cross-border developments which are reported till 2035 in these documents. The general trend in cross border capacity development shows significant (2-5 years) delays in the realisation of the cross-border capacities across Europe, so we considered, that proposed projects till 2035 will be realised till

2040. The reference NTC values are included in the online annex for each direction for the two corner years of 2030 and 2040.

Analytical approach

In general, I find the approach too static, only using a dispatch and no investment model and not showing any counterfactuals for other integration options.

Answer: Thanks for the comment. In the revised version we have switched to use the investment module of the power market model to overcome this critique and to enable more robust, less static assessment of the future development. The results are updated, and we present a range of result for a wide range of outcome based on the new modelled investment ranges. We believe it also helped a lot in overcoming the too deterministic approach used in the first submission.

In the results section, I am missing information on the market value of renewables, specifically PV. Ideally, this should be distinguished by “optimized” and “bifacial vertical”. The introduction mentions the cannibalization of renewables but fails to show this effect in the analysis.

Answer: Thanks for the suggestion, this covers an important aspect of the future power sector development of the EU, as we see strong market value reduction in the case of the PV technology. The market value changes are now provided in the new Figure 4, and comparison is provided to the study of Bernath et al (2021) after the figure. We have similar market value reduction to the cited study in the Low PV deployment case (around 50%), but we have a much stronger market value reduction effect in the high PV case, mainly due to the much stronger assumption on PV deployment. It needs to be emphasised, that the model automatically considers this market value reduction of PV in its optimization and dispatch.

See Figure 5 for details and the explanation following Figure 5 on the Market value development.

Suggested improvements

As mentioned, the analysis is too static and would gain significantly from testing counterfactuals. See also my comments on data and methodology.

Answer: see our answers given in the previous questions, and the description of how the investment module is used in the revision.

The abstract should be improved. Currently, it does not become clear what the contribution of the paper is and what novel/diversified installations have been considered.

Answer: The abstract was changed to better reflect the contribution of the paper

I am also strongly arguing in favour of publishing the methodology code under an open-source license and (or at minimum) the resulting data for further use in energy system modelling studies.

Answer: We have assembled the dataset with the resulting data of all modelled scenarios, sensitivity cases, for the years 2030 and 2040, and provide it country by country in an online annex accessible to

all readers for further use.

The quality of the graphs needs to be improved. Most of them are illegible and should be provided as vector graphics or at least with a significantly higher resolution.

Answer: The graphs have been completely reworked, to make the more legible and using the same style as well as to reflect the extended model approach.

Clarity and context

The paper is written in a very clear and accessible way. It is well structured and easy to follow, and the line of arguments is well presented. I do recommend a thorough proofread to eliminate the few existing typos.

Answer: The text was checked for the typos by the authors as well as a native speaker.

References

While the paper references previous literature to the extent serving its results, I am missing some aspects: (1) alternative instruments like two-sided CfDs (cf. e.g., <https://doi.org/10.1038/s41560-023-01401-w>), (2) alternative integration options like flexibility from electrification (cf. e.g., <https://doi.org/10.1016/j.energy.2023.127832>), and (3) the effect of system integration on market values (cf. e.g., <https://doi.org/10.1016/j.apenergy.2020.115985>).

Answer: The references have been extended with a number of new studies including the recommended ones.

Reviewer #3 (Remarks to the Author)

Comments to the Author

This paper proposed “Disruptive deployment of PV technology unlocking the full potential of solar energy in the merit order”. The presentation of the submitted manuscript and the proposed approach requires major improvements of the paper. This reviewer would like to see the effort of the authors to revise the paper and addressing the following questions:

1) Clarity is required in the abstract on the contribution of the proposed work.

Answer: The abstract has been refocused on the novelty of the contribution to present day most current energy and climate policy discussion.

2) The introduction needs to be revisited. It is quite elaborate and lacks direction with a clear research gap. There is repetition of information at many places. The contribution of the work must be thoroughly rewritten with more clarity.

Answer: The introduction has been streamlined and refocused to avoid repetitions and divergent directions, to put more emphasis on the core message on the needed renewable technology deployment change and its impact.

3) What is the major contribution of the paper? More and clear presentation of the paper contribution in abstract and introduction is necessary.

Answer:

The paper significantly contributes by addressing the root cause of concerning market signals, such as volatile and often negative prices associated with PV output. It proposes a scalable PV technology option, particularly emphasizing bifacial panels, suitable for rapid deployment in various settings including freestanding arrays, agrivoltaics setups, and roadside integration. Furthermore, the study integrates these findings into a comprehensive techno-economic model with spatial and temporal resolution. This model considers seasonal variations of different technologies and evaluates their impacts on both markets and climate performance.

This work was developed under a multi-disciplinary approach through a collaborative effort involving multiple research groups. These include a team specializing in novel PV technology testing measurements, an economics modelling team that developed various EU energy models and their interactions between conventional and renewable technologies, and a market research analyst affiliated with an electricity trade company.

The abstract and introduction now provide a clearer presentation of these contributions.

4) The novelty of the work must be clearly addressed and discussed, compare your research with existing research findings and highlight novelty, (compare your work with existing research findings and highlight novelty).

Answer: By the more thorough comparison with the most recent research findings and completing the techno-economic analysis on the impacts of the new PV production patterns in mass solar production and in the overall electricity market we are convinced that the presented novel application has the potential to create the necessary market dynamism to reach the targeted climate neutral electricity portfolio until the other market mechanisms and net zero & storage technologies presently in a lower TRL level can scale up. The literature overview was also widened in this revision process to more than 50 references.

5) Why did the authors focus the paper on disruptive deployment of PV technology?

Answer: In reaching the European Green Deal investments in solar PV has been a success story: the installed capacities have overpassed the targets. However, the mass integration of PV has brought up price fluctuation problems in the electricity market that will undermine the investors' confidence, if it is not dealt with in a timely policy action. The EU market has seen similar disruption in the PV installation during the 2013-14 when the dominant support policy regimes (the feed-in-tariffs) were adapted too late, and the sudden stop&go policy made the investors uncertain. The novel PV deployment option analysed in the paper could assist to maintain the investment level and give time to the other flexibility and storage solutions could scale up.

6) The methodology is not very clear, the author needs to clarify and justify.

Answer: During the revision the applied model has been complemented with an investment model and the description of the assumptions, the used input data (as an annex) has been added.

7) The supply and demand side model is not clear, authors need to justified.

Answer: The country-specific consumption data is based on the Impact Assessment for FITfor55 packages of the European Commission (https://energy.ec.europa.eu/data-and-analysis/energy-modelling/policy-scenarios-delivering-european-green-deal_en). The supply side and the capacity expansion part of the model is now more detailed now in the methodology section.

8) In case studies, some assumptions need to be clarified and justified.

Answer: The modelled portfolio and the assumptions now have been completed. The main assumption on inputs (such as fuel and CO2 prices) are introduced, the more detailed technology and country specific input data (such as emission factors, country specific net transfer capacity values, technology cost data) and assumptions are now included in the online annex for further clarity and use.

9) The result portion needs to be re-written from scratch, to bring about flow of thought.

Answer: In the revised version, the investment module determines PV and wind investment, which changed the results. With these new results all the figures and tables are re-assembled, and text of the result and conclusion sections are also re-written to have a better flow of thoughts.

10) More updated references for the years 2022 and 2023 can be included for literature review.

Answer: The reference list has been extensively extended with the most recent relevant works, our reference list is now extended to more than 50 references.

11) Conclusion section is missing some perspective related to the future research work,

Answer: The future research directions on the future policy, market and technology aspects has been identified in the discussion section, including the implications of hydrogen production, many other types of other energy reserves heat pumps and also the wind-PV complementarities.

12) Please improve the quality of figures

Answer: The figures have been completely reworked, to make the more legible and using the same style as well as to reflect the extended model approach.

13) English language should be carefully checked and carefully check paper for language typos.

Answer: The text was checked for the typos by the authors as well as a native speaker.

REVIEWER COMMENTS

Reviewer #1 (Remarks to the Author):

The authors have dealt with the comments raised, although the literature review could be more thorough. Using as references papers that are still under review is unusual.

Reviewer #2 (Remarks to the Author):

Thank you for addressing my previous comments. I still have a few remaining questions/comments.

~ In General: I still think the analysis is too static. You still claim 2040 to be "short-term" when this is still 15 years from now and 10 years before the EU wants to reach climate neutrality after the recommended -90% in 2040. I, therefore, see a problem in that the analysis does not include further flexibility options in the investment model. The paper can in this way only contribute insights from a static comparison with different shares of bifacial PV everything else being constant. This will clearly affect prices and market values. The paper can still be valuable on this very specific aspect but you need to very clearly state these shortcomings or, alternatively, direct the focus more on a 2030 scenario. Otherwise, you suggest findings that - at least in my opinion - you cannot justify with the experimental setup used.

~ Figure 6: How do you explain the increase in consumption in a lot of countries when the total generated electricity in 2040 (Figure 4) is almost identical in the two extreme scenarios? I could not find this in the text.

~ There seems to be missing some text in the section "Alternative deployment options in the EU" as it starts mid-sentence.

~ I still strongly recommend publishing the code used for the paper open-source on GitHub for reproducibility (not just the data).

Reviewer #3 (Remarks to the Author):

Accept in current form

REVIEWER COMMENTS

The authors would like to thank the referees their work and constructive recommendations and comments to the manuscript.

Reviewer #1 (Remarks to the Author):

The authors have dealt with the comments raised, although the literature review could be more thorough. Using as references papers that are still under review is unusual.

We have inserted new references and the two references that were quoted as "uder review" has been accepted, so we could make actual references to these two in the final reference list.

Reviewer #2 (Remarks to the Author):

Thank you for addressing my previous comments. I still have a few remaining questions/comments.

~ In General: I still think the analysis is too static. You still claim 2040 to be "short-term" when this is still 15 years from now and 10 years before the EU wants to reach climate neutrality after the recommended -90% in 2040. I, therefore, see a problem in that the analysis does not include further flexibility options in the investment model. The paper can in this way only contribute insights from a static comparison with different shares of bifacial PV everything else being constant. This will clearly affect prices and market values. The paper can still be valuable on this very specific aspect but you need to very clearly state these shortcomings or, alternatively, direct the focus more on a 2030 scenario. Otherwise, you suggest findings that - at least in my opinion - you cannot justify with the experimental setup used.

We have placed an extended shortcomings sections in the conclusions part as follows:

The analysis in this paper presents a limited comparison of the impact of the modelled higher penetration of bifacial PV systems, assuming that all other important variables remain unaffected. This means the other flexibility options (e.g. increasing transfer capacities between countries, higher level of storage and demand side options) are kept at their reference pathway, which means the dynamic interactions with these options are kept limited in the analysis. This is a strong assumption, as the modelling is applied for the next 15 years. As these other options would affect wholesale prices and market values of production, the comparison remains rather static, but it still indicates the range of potential contribution of the bifacial technology to the power sector transformation.

~ Figure 6: How do you explain the increase in consumption in a lot of countries when the total generated electricity in 2040 (Figure 4) is almost identical in the two extreme scenarios? I could not find this in the text.

In Figure 4, the total yearly generation difference between the two extreme scenarios in 2040 is 37 TWh (3393 TWh in the 100-0% scenario and 3431 TWh in the 50-50% scenario). In Figure 6, the highest changes in consumption levels in the two most extreme scenarios are in Germany (3.8 TWh), Spain (3.7 TWh), and France (3.5 TWh). If we add all the EU27

countries, the total consumption change is 22.4 TWh. The difference between the 37 TWh (from Figure 4) and the 22.4 TWh is due to the net import of the EU27, which also changes between the scenarios.

We have placed this explanatory text into the paper:

The change in production level in the EU is smaller than the change in consumption, due to the increased net export of the EU to outside regions in the bi-facial heavy scenarios.

~ There seems to be missing some text in the section "Alternative deployment options in the EU" as it starts mid-sentence.

The missing start of the sentence was inserted.

~ I still strongly recommend publishing the code used for the paper open-source on GitHub for reproducibility (not just the data).

We placed the detailed mathematical equations of the model on the objective function and the constraints in Github (link). We believe the mathematical equations are more comprehensible for majority of experts than a specific code, as it alone would still be a black box for the majority of readers not familiar with the specific modelling language (in this case Julia). In this way the model equations could be rewritten to any modelling languages the user is familiar with.

Reviewer #3 (Remarks to the Author):

Accept in current form

REVIEWERS' COMMENTS

Reviewer #2 (Remarks to the Author):

Thank you for considering my comments. I have no further ones and recommend the acceptance of the manuscript.

Reviewer #2 (Remarks on code availability):

The links shown here did not work but I was able to access the repository via the GitHub profile - it should be double-checked.

I still think it would be valuable to also provide the full code but I am also happy for the documentation of the used equations.